# Pyrrolizidine Alkaloids: Biosynthesis, Biological Activities and Occurrence in Crop Plants

**DOI:** 10.3390/molecules24030498

**Published:** 2019-01-30

**Authors:** Sebastian Schramm, Nikolai Köhler, Wilfried Rozhon

**Affiliations:** Biotechnology of Horticultural Crops, TUM School of Life Sciences Weihenstephan, Technical University of Munich, Liesel-Beckmann-Straße 1, 85354 Freising, Germany; seb.schramm@tum.de (S.S.); nikolai.koehler@tum.de (N.K.)

**Keywords:** *Borago officinalis*, *Crassocephalum*, Copper-dependent diamine oxidase, *Gynura bicolor*, Homospermidine synthase, *Lolium perenne*, Necic acids, Necine bases, Pyrrolizidine alkaloid biosynthesis, Senecionine

## Abstract

Pyrrolizidine alkaloids (PAs) are heterocyclic secondary metabolites with a typical pyrrolizidine motif predominantly produced by plants as defense chemicals against herbivores. They display a wide structural diversity and occur in a vast number of species with novel structures and occurrences continuously being discovered. These alkaloids exhibit strong hepatotoxic, genotoxic, cytotoxic, tumorigenic, and neurotoxic activities, and thereby pose a serious threat to the health of humans since they are known contaminants of foods including grain, milk, honey, and eggs, as well as plant derived pharmaceuticals and food supplements. Livestock and fodder can be affected due to PA-containing plants on pastures and fields. Despite their importance as toxic contaminants of agricultural products, there is limited knowledge about their biosynthesis. While the intermediates were well defined by feeding experiments, only one enzyme involved in PA biosynthesis has been characterized so far, the homospermidine synthase catalyzing the first committed step in PA biosynthesis. This review gives an overview about structural diversity of PAs, biosynthetic pathways of necine base, and necic acid formation and how PA accumulation is regulated. Furthermore, we discuss their role in plant ecology and their modes of toxicity towards humans and animals. Finally, several examples of PA-producing crop plants are discussed.

## 1. Introduction

Pyrrolizidine alkaloids (PAs) are heterocyclic organic compounds synthesized by plants that are thought to act as defense compounds against herbivores [1]. Estimates indicate that approximately 6.000 plant species worldwide, representing 3% of all flowering plants, produce these secondary metabolites. In particular, members of the Asteraceae, Boraginaceae, Heliotropiaceae, Apocynaceae, and some genera of the Orchidaceae and the Fabaceae are PA producers [2]. Reported concentrations vary greatly, from trace amounts to up to 19% dry weight, and are considered to be dependent on a number of factors including the developmental stage, tissue type, environmental conditions, and extraction procedures [3].

PAs consist of a necine base esterified with a necic acid. The necine base typically includes pyrrolizidine, a bicyclic aliphatic hydrocarbon consisting of two fused five-membered rings with a nitrogen at the bridgehead [4] (Figure 1). Loline alkaloids may be formally considered as PAs since they also possess a pyrrolizidine system, although it contains an ether bridge linking carbon 2 (C-2) and carbon 7 (C-7). While *stricto sensu* PAs are exclusively formed in plants, lolines are synthesized by endophytic fungal symbionts of the genus *Epichloë* [5]. In addition, their biosynthesis is distinct from PAs [5,6,7]. Thus, lolines will be discussed only peripherally in this review.

The fused bicyclic system of PAs resembles indolizidine and quinolizidine alkaloids, which contain a five and a six-membered ring or two six-membered rings, respectively [8] (Figure 1). Tropane and granatane alkaloids also show a similar structure consisting of a five and a six-membered ring or two six-membered rings, respectively [9]. However, in contrast to necine bases, the rings of tropane and granatane alkaloids are bridged rather than fused. While several tropane and quinolizidine alkaloids including atropine (the racemic mixture of (±)-hyoscyamine) and sparteine are used in medicine [9], PAs are mainly known for their hepatotoxic and potentially carcinogenic properties [10]. Nevertheless, some PAs show interesting pharmacological properties that are currently under investigation (see Section 5.3) [10,11]. While tropane and quinolizidine alkaloids are usually present in plants in their free forms, PAs are mainly present as *N*-oxides (Figure 1), which are highly water-soluble and considered less toxic than the free PAs.

## 2. Structural Diversity of Pyrrolizidine Alkaloids

Within the combination of a set of necine bases (Figure 2 and Figure 3) and a considerable number of necic acids (Figure 4), an enormous structural diversity of PAs can be obtained. This is further amplified by modifications, including *N*-oxidation of the tertiary nitrogen of the necine base, hydroxylation of the necine base and/or the necic acid, and acetylation of hydroxy groups of the acid moiety. Thus, it is not surprising that several hundreds of different PAs have already been identified and each year new variants are described. 

### 2.1. Diversity of Necine Bases

In addition to the pyrrolizidine ring system most necine bases possess a hydroxymethyl group at position 1 (Figure 2), which is a consequence of the biosynthetic pathway (see Section 3.1). Since 1-hydroxymethylpyrrolizidine contains two chiral centers, carbons C-1 and C-8, in total four compounds exist: The enantiomers (-)/(+)-trachelanthamidine and (-)/(+)-isoretronecanole (Figure 2B). Among them, (-)-trachelanthamidine and (-)-isoretronecanole are most frequently found, for instance, as the necine base of trachelanthamine (Figure 5C) and the nervosines [12] (Figure 5E), respectively. Examples for PAs containing (+)-trachelanthamidine and (+)-isoretronecanole are acetyllaburnine [13] (Figure 5G) and madhumidine A [14], respectively. The most frequent modification of saturated necine bases is hydroxylation at C-7. However, the positions C-2 and C-6 are also occasionally hydroxylated. Necine bases like (-)-platynecine, possessing hydroxy groups on C-7 and C-8, and (-)-rosmarinecine containing hydroxy groups on C-2, C7 and C-9 (Figure 2), are often esterified by dicarboxylic necic acids to form macrocyclic PAs like platyphylline and rosmarinine (Figure 5A) [15]. In general, saturated PAs are considered as non-toxic [16].

Most PAs contain a necine base possessing a double bond between C-1 and C-2 (Figure 2). Introducing that double bond eliminates the chiral centre at C-1, thus leaving only the stereocentre at C-8. Consequently, only two forms, (-)/(+)-supinidine, exist of the C-9 monohydroxalated derivatives and four, (-)/(+)-retronecine and (-)/(+)-heliotridine, of the C-7 and C-9 dihydroxylated compounds. Among them (+)-retronecine is the most frequently observed necine base in PAs.

In addition to the saturated and desaturated bases discussed above, necine bases of the otonecine type also exist. Otonecine is not a genuine bicyclus, but may act as such due to transannular interactions of the keto group and the tertiary amine (Figure 3A) [17]. These interactions are also likely for the reason that otonecine-type PAs are present in plants as free bases rather than *N*-oxides. 

There are also several necine bases with unusual structures (Figure 2D). One of them is 1-aminopyrrolizidine, wherein the hydroxymethyl group is replaced by an amino group. This unusual necine base is found for instance in laburnamine, an alkaloid present in trace amounts in *Laburnum anagyroides* [18]. From the leaves of *Ehretia asperia,* ehretinine was isolated, which is very unusual since the 7-hydroxy group of its necine base, (1*R*,7*S*)-7-methylhexahydro-1*H*-pyrrolizin-1-ol, is esterified with 4-methylbenzoic acid and the typical hydroxymethyl group on C-1 is replaced by a methyl residue [19]. Similarly, *Senecio polypodioides* contains, besides sarracine *N*-oxide, also 7β-angeloyloxy-1-methylene-8α-pyrrolizidine, a PA with a methylene group instead of the typical hydroxymethyl residue on its necine base. The 7-hydroxy group of this PA is esterified with angelic acid [20]. In *Echium glomeratum*, PAs with a tricyclic ring were found. The 9-hydroxy group of the necine base was found to be esterified with angelic acid [21]. Another example is the necine base of tussilagine from *Tussilago farfara*, which possesses a carboxy group instead of the typical hydroxymethyl group (Figure 5H) [22].

### 2.2. Diversity of Necic Acids

While necine bases share a common structure, the necic acids show broad structural diversity. Some, particularly the smaller and simpler ones, are typical metabolites of plant metabolism, while others, particularly the monocarboxylic acids of the trachelanthic acid type and the dicarboxylic acids (Figure 4) are formed in specific, complex pathways.

Acetic acid (Figure 4A) is frequently observed in simple PAs, for instance 7-acetylretronecine present in *Onosma arenaria* [23] and acetyllaburnine present in *Vanda*, a genus of the Orchidaceae [13,24] (Figure 5G). Acetic acid may also esterify the second hydroxy group of the necine base in triangularine and lycopsymine-type PAs, such as 7-acetyl-9-sarracinoylretronecine present in *Alkanna tuberculata* [25] (Figure 5B) and uplandicine found in pollen of *Echium vulgare* (Figure 5C). In addition, acetic acid also frequently esterifies hydroxy groups of other necic acids in more complex PAs, for instance florosenine [26], ligularidine [27] (Figure 3B), or acetylerucifoline *N*-oxide [28]. In contrast to the frequently observed acetic acid, lactic acid has, so far, only been found in lactoidine, a PA of *Cynoglossum furcatum* [29].

C_5_ acids of the tiglic acid type (Figure 4A) are characteristic for the triangularine group of PAs (Figure 5B). They may esterify one or two hydroxy groups of the necine base. In the former case they may appear together with acetic acid or more complex necic acids, particularly branched C_7_ acids, which is seen for instance in the PAs echimidine [30] and heliosupine [31] (Figure 5F). In addition to esterifying necine bases directly, C_5_ acids may also esterify hydroxy groups of other necic acids. Examples are scorpioidine, a PA of *Myosotis scorpioides* [32] (Figure 5F), and anadoline, a PA of *Symphytum orientale* [33,34]. Latifolic acid [35,36,37] and the closely related hackelic acid [38] are examples of cyclic C_7_ acids.

Aromatic systems are rarely present in necic acids except in PAs found in the Orchidaceae. Many of them, for instance benzoic acid, salicylic acid and *p*-coumaric acid, are simple aromatic acids present as primary or secondary metabolites in most plant species. However, some aromatic necine bases, particularly those found in the genera *Phalaenopsis* and *Liparis*, show a very complex structure, for instance the phalaenopsines [39,40] and the nervosines [41] (Figure 4B and Figure 5E).

The dicarboxylic necic acids (Figure 4C) are a particularly interesting group because they form macrocyclic PAs, which are considered to be the most toxic. Necic acids of the monocrotalic acid type are a relatively small group; they form 11-membered rings. In contrast, senecic acid-like necic acids typically form 12-membered rings and represent a large group. The considerable diversity is obtained by modification of the senecic acid core structure by a number of reactions (Section 3.2.4). Interestingly, a few among them contain chlorine (Figure 3B and Figure 4C), a modification rarely observed in plant metabolites.

### 2.3. Linkage Patterns of Necine Bases with Necic Acids

Based on the combination of necine bases and necic acids and their linkage patterns the PAs have been classified into five groups [42]. The first and largest group are senecionine-like PAs, which consist of necine bases of the retronecine (Figure 2C), platynecine, rosmarinecine (Figure 2B), or otonecine-type (Figure 3) and typically branched C_10_ dicarboxylic necic acids (Figure 4C) derived from two molecules of l-isoleucine (see Section 3.2.4), which together form 12-membered macrocyclic rings. An exception is the small sub-group of nemorensine-like PAs [43], which form 13-membered macrocycles (Figure 5A). Typically, the necine bases are esterified at their C-7 and C-9 hydroxy groups. PAs of this type are mainly found in the tribe Senecioneae and family Fabaceae [44]. 

The second group is represented by triangularine-type PAs, which are open-chain mono- or diesters of necine bases with the C_5_ acids tiglic, angelic, senecioic, and sarracinic acid (Figure 5B). These PAs are mainly present in Senecioneae and Boraginaceae [44]. 

The third type, the lycopsamine-like PAs are mainly found in Boraginaceae and Eupatorieae [44]. This type possesses branched C_7_ necic acids esterifying the C-9 hydroxy group (Figure 5C). A number of PAs represent a combination of group 2 and 3 since they also possess a C_5_ acid residue in addition to a C_7_ necic acid. The C_5_ acid residue can either be linked directly with the necine base or attached to a hydroxy group of the C_7_ acid (Figure 5F). 

The fourth group are the 11-membered macrocyclic PAs of the monocrotaline type. Similar to senecionine-like PAs the hydroxy groups of C-7 and C-9 are esterified with dicarboxylic necic acids (Figure 5D). This group is found predominantly in Fabaceae [44]. 

Phalaenopsine and ipanguline-type PAs represent the fifth group, which is characterized by the presence of an aromatic acid (Figure 4B), esterifying the usually saturated necine base (Figure 5E). The acidic compound shows a high structural diversity and includes simple aromatic acids like benzoic, salicylic and *p*-coumaric acid, but also very complex ones like nervogenic acid. This is the only group of PAs that are frequently glycosylated. Members of this group are found in the Orchidaceae, Convolvulaceae, and in a few representatives of other tribes including the Boraginaceae [44].

In addition to these five groups, there are also very simple PAs consisting only of the necine base and a small acid residue, particularly acetate, as illustrated by the examples shown in Figure 5G. A number of PAs show unusual linkage patterns distinct from that of the five groups discussed above. In madurensine the hydroxy group of C-9 is bridged by the dicarboxylic acid intergerrinecic acid with a hydroxy group placed at C-6 rather than the usual C-7 hydroxyl [45]. This leads to a 13-membered macrocyclic ring (Figure 5H). The structure of laburnamine [46] matches that of PAs of the triangularine type. However, since its necine base (1*S*,8*R*)-1-aminopyrrolizidine (Figure 2D) possesses an amino group instead of the hydroxy group on C-9, a reaction with isovaleric acid yields an amide rather than an ester bond (Figure 5G). Tussilagine, a PA of *Tussilago farfara* (coltsfoot), is very special since its necine base possesses, instead of the typical hydroxymethyl residue, a carboxy group on C-1, which is esterified with methanol [22]. Anhydroplatynecine is devoid of any necic acid and the C-7 and C-9 hydroxy groups of platynecine (Figure 2B) instead combine together via an ether bridge. However, anhydroplatynecine is likely not a naturally-occurring PA, but is rather formed by heating of platynecine containing PAs during isolation [47]. Finally, it is worth mentioning that several plant species also contain unmodified necine bases in their free form or as *N*-oxides [48].

### 2.4. Modification and Conjugation of Pyrrolizidine Alkaloids

As discussed above, the astonishing diversity of PAs is achieved by hydroxylation and desaturation of necine bases and necic acids and their combination to PAs. Complete PAs might also be modified by hydroxylation, desaturation and epoxidation. The latter may be further metabolized to a diol or a chlorine-containing PA (see Section 3.2.4). In addition to these generally irreversible modifications, PAs can also be reversibly modified. By far the most frequently observed modification of this type is *N*-oxidation (Figure 1). In plants, the major fraction of PAs is present as *N*-oxides. Exceptions include seeds of several *Crotalaria* species [49] and leaves of *Crassocephalum crepidioides*, wherein the majority of the PAs are present in their basic form [50], and shoots of jacobine-chemotype plants of *Senecio jacobaea*, in which up to 50% might be present as tertiary PAs [51]. *N*-oxidation of the tertiary amine nitrogen changes the properties of a PA significantly. In contrast to basic tertiary amines, which are positively charged under physiological conditions, amine *N*-oxides are neutral and behave like very polar, highly water-soluble, salt-like compounds that are thought to be membrane impermeable. These characteristics might be important for their role in transport and storage of PAs. Accordingly, it was shown that PA transporters in membranes of plant cells have a higher affinity for PA *N*-oxides than for the tertiary amines [52]. 

In addition to *N*-oxidation, a number of PAs are also acetylated, particularly at hydroxy groups of the necic acid moiety. Examples are 7-acetylscorpionidine, the 7-*O*-acetylation product of scorpionidine (Figure 5F), and the otonecine-type PAs florosenine and ligularidine (Figure 3B), which are acetylated forms of otosenine and petasitenine, respectively. 

While glycosylation is frequent among secondary metabolites, modifications of that type are rarely observed for PAs. Only among the PAs with aromatic necic acids some examples are known. They include thesinine-4′-*O*-α-l-glucoside present at high levels in borage seeds [53] (Section 6.1), thesinine-4’-*O*-α-l-rhamnoside found in *Lolium* species [54] (Section 6.4) and nervone PAs isolated from *Liparis nervosa* [41] (Figure 5E).

Other modifications are rarely seen in PAs.

## 3. Biosynthesis of Pyrrolizidine Alkaloids

Attempts at deciphering PA biosynthesis (Figure 6) date back to the early 1960s, when Nowacki and Byerrum performed their first feeding experiments with radiolabeled precursors [55,56]. Later, this work was continued by others, mainly the groups of Robins and Crout. Robins also introduced labeling with stable isotopes, particularly ^13^C, ^2^H and ^15^N, and subsequent analysis by NMR spectroscopy for analysis of PA biosynthesis [57,58,59]. This technique provided detailed information about the fate of single C and H atoms during biosynthesis of the necine bases and necic acids. In the late 1990s, the first biosynthetic enzyme, homospermidine synthase, catalyzing the first committed step in PA biosynthesis, was identified [2]. Its analysis in different plant species provided interesting data about PA evolution, or more precisely, homospermidine biosynthesis.

### 3.1. Biosynthesis of Necine Bases

Feeding of *Crotalaria spectabilis* plants, which produce monocrotaline, with ^14^C-labelled precursors showed that [^14^C]-ornithine was efficiently incorporated into monocrotaline, particularly into its necine base retronecine [56]. Studies with *Senecio isatideus* [60] and *Senecio douglasii* [61] confirmed that [^14^C]-ornithine is mainly incorporated into the necine base. Degradation studies in the latter study showed that approximately 25% of the incorporated radioactivity was present in carbon C-9 of the retronecine unit irrespective of whether [2-^14^C]-ornithine or [5-^14^C]-ornithine were fed, indicating that C-2 and C-5 of ornithine become equivalent during biosynthesis (at least for biosynthesis of the right-handed ring) and suggesting 1,4-diaminobutane (putrescine) as a symmetrical intermediate. Indeed, feeding of [1,4-^14^C_2_]-putrescine again yielded retronecine bearing approximately 25% of the radioactivity on C-9. Using *Senecio magnificus,* it was shown that arginine is also selectively incorporated into the necine base part of senecionine [62]. Subsequently, Robins and Sweeney compared the incorporation efficiency of several compounds using *Senecio isatideus* and found that putrescine, spermidine, and spermine were more efficiently incorporated than arginine and ornithine [63] and that the two latter were only effectively incorporated when present in the L configuration [64]. 

While these experiments established l-arginine and l-ornithine as precursors and putrescine as an intermediate, they did not allow a more comprehensive investigation since the fate of the individual C and H atoms could not be followed during biosynthesis. This problem was overcome by introduction of stable isotope labeled precursors and NMR analysis of the obtained products in combination with an improved plant feeding technique. Previously, plants were mainly fed hydroponically or as cut shoots, which resulted in incorporation rates significantly below 1%. In contrast, by absorption of aqueous solutions of the precursors directly into the xylems of freshly rooted cuttings through stem punctures, incorporation rates of up to 5% could be obtained [63]. Feeding of both [1,4-^13^C_2_]-putrescine and [1-^13^C]-putrescine gave enriched ^13^C signals for C-3, C-5, C-8 and C-9 (Figure 7A), confirming that both rings originate from putrescine [57,65].

Because approximately 1.1% of the natural occurring carbon is the ^13^C isotope, a ^13^C label is easily obscured, particularly if the incorporation efficiency is moderate to low. Thus, further studies made use of double labeled precursors, where the ^13^C-^13^C double label can be sensitively detected by ^13^C-NMR spectroscopy as doublet around the natural abundance signal. Feeding of [2,3-^13^C_2_]-putrescine gave a pair of doublets for C-1 and C-2 with a coupling constant *J* of 34 Hz and a second doublet pair for C-6 and C-7 with a coupling constant *J* of 70 Hz [65] (Figure 7B). Finally, feeding *Senecio isatideus* with [1,2-^13^C_2_]-putrescine gave rise to four pairs of doublets, namely C-1/C-9, C-2/C-3, C-5/C6 and C-7/C-8, with four different coupling constants (Figure 7C). To investigate which C-N bond remains intact during retronecine biosynthesis [1-^15^N,1-^13^C], double labeled putrescine was synthesized and fed to *Senecio vulgaris* [66,67] or *Senecio isatideus* [58,65]. Analysis of the necine base showed an equal amount of retronecine with a ^15^N and a ^13^C label at C-3 and retronecine with a ^15^N and a ^13^C label at C-5 (Figure 7D). The observation that both variants appeared at the same level confirmed the presence of a symmetric C_4_-N-C_4_ compound in retronecine biosynthesis. A similar series of experiments was also performed using *Senecio pleistocephalus*, which forms rosmarinine as the sole PA. Rosamarinine consists of senecic acid and the saturated necine base rosmarinecine, which has, in addition to the typical 7 and 9-hydroxy groups, an additional hydroxy group on C-2 (Figure 2B). Additionally, within this experimental system, the same result was obtained that a symmetric C_4_-N-C_4_ intermediate is involved in necine base biosynthesis [68]. 

#### 3.1.1. Homospermidine Synthase

The most obvious candidate for the symmetric C_4_-N-C_4_ intermediate was homospermidine, which is known to be present in a number of plant species, particularly in such producing PAs. Initial experiments with radiolabeled homospermidine showed that this compound was incorporated into the necine base part of retrorsine in feeding experiments with *Senecio isatideus* [69]. To investigate whether the C_4_-N-C_4_ unit stays intact during biosynthesis ^13^C labels were placed on the most distal carbons (Figure 7E). Feeding of [1,9-^13^C_2_]-homospermidine to *Senecio pleistocarpus* revealed enrichment of the label on C-8 and C-9 of the isolated retronecine [70], confirming that homospermidine is incorporated intact into necine bases. 

At the same time, Bötcher et al. partially purified an enzyme with homospermidine synthase (HSS) activity from root cultures of *Eupatorium cannabinum* [71]. Enzymatic assays were performed by addition of [^14^C]-labeled putrescine to the partially purified enzyme. The pH optimum was 9 and the enzymatic reaction was strictly dependent on the presence of NAD^+^. The enzyme showed high selectivity for putrescine and NAD^+^, and accepted neither NADP^+^ as co-substrate nor any of the other tested amines including 1,3-diaminopropane, cadaverine, and pyrroline as substrate. The Michaelis-Menten constants (K_M_) for putrescine and NAD^+^ were 13.5 µM and 3 µM, respectively. In contrast to NAD^+^, its reduced form, NADH, acted even at low concentrations (inhibition constant K_i_: 2 µM) as a strong inhibitor indicating that the NADH formed in the first step of the reaction, the formation of an imine intermediate, remains bound to the enzyme and serves as electron donor for the second step, the reduction of the imine intermediate to the secondary amine. The enzymatic activity was also inhibited by 1,3-diaminopropane, spermidine, and homospermidine with K_i_ values of 6.3, 94 and 950 µM, respectively. While it was believed that the partially purified enzyme utilized two molecules of putrescine for formation of one molecule of homospermidine, later work with HSS purified to homogeneity showed that the enzyme instead uses putrescine and spermidine to produce homospermidine and 1,3-diaminopropane as by-products [2]. The reason for the initially wrong conclusion was that spermidine was added at all steps during enzyme purification since it was found that this compound preserves enzyme activity. Consequently, spermidine was also present in the partially purified enzyme and thus in the enzymatic reaction at sufficient concentrations allowing transfer of the radiolabeled putrescine moiety to spermidine. Thus, the first step of PA biosynthesis is transfer of the 1,4-diaminobutan part of spermidine to a specific lysine residue of HSS accompanied by release of 1,3-diaminopropane and reduction of NAD^+^ to NADH, which remains bound to HSS (Figure 8). Subsequently, putrescine reacts with the HSS-bound 1,4-diaminobutane moiety under formation of an imine intermediate and regeneration of the lysine-NH_2_. Finally, the imine is reduced by the HSS-bound NADH to homospermidine and released from the regenerated HSS/NAD^+^ complex [72].

Purification of HSS from *Senecio vernalis* to homogeneity allowed identification of the protein and corresponding gene. After protease treatment, four fragments were obtained that were microsequenced to give short peptide sequences. Database searches revealed close homology to deoxyhypusine synthase (DHS) [2]. DHS catalyzes the NAD^+^-dependent transfer of an amino-butyl moiety from spermidine to a specific lysine side chain of the precursor for the eukaryotic initiation factor 5A (eIF5A), forming the amino acid deoxyhypusine. This reaction is one of the most specific post-translational modifications known [73,74] and similar to the reaction proposed for HSS except that elF5A is replaced by spermidine (Figure 8). Subsequent PCR with redundant primers yielded the first sequences, which were completed by 3´ and 5´-RACE allowing subsequent for a cloning of the complete *Senecio vernalis* HSS coding sequence. Heterologous expression of the cloned gene as His_(6)_-tagged fusion protein in *E. coli* and in vitro enzymatic assays confirmed that the enzyme for homospermidine synthesis was obtained and that putrescine and spermidine are required as substrates. With respect to that, it is worth mentioning that this may also explain the considerable incorporation of radiolabeled spermidine into homospermidine previously observed in feeding experiments using *Senecio isatideus* [63]. 

Since HSS catalyzes the first committed step in PA biosynthesis, thereby linking primary metabolism with PA biosynthesis, it is also interesting to study its evolution in order to deduce how PA biosynthesis was established in plants [72]. Due to 70 to 90% sequence homology—depending on the species—it was suggested that HSS evolved through gene duplication from DHS [75]. HSS has been recruited from DHS independently in different plant families at least eight times: Once in the Apocynaceae, Boraginaceae, Convolvulaceae, Fabaceae, Orchidaceae, and Poaceae, and twice in the Asteraceae [76,77,78]. The major difference lies in the biochemical properties of the two enzymes. DHS accepts eIF5A (lys) as well as putrescine, though with far lower efficiency, as substrate and can therefore catalyze the formation of both eIF5A(dhp) and homospermidine, the latter at a very low rate. In contrast, HSS has lost the ability to bind eIF5A, hence can only produce homospermidine [2,79,80]. This change in substrate specificity was possible because binding of putrescine occurs within the active site of DHS, while binding of eIF5A(lys) happens at the surface of the enzyme [75]. Reimann et al. [78] compared the rates of non-synonymous to synonymous mutations in HSS and DHS and found higher rates in HSS, suggesting higher selection pressure on DHS. The same study also found that it is not possible to distinguish between DHS and HSS solely by sequence data since there are no differing characteristic patterns. Despite the similar biochemical properties and sequences, the expression levels of both genes clearly differ. A study of the expression patterns of HSS and DHS in *Senecio vernalis* (Asteraceae, Senecioneae) revealed that DHS is expressed in all plant tissues in an almost constant manner throughout plant development. In contrast, HSS expression was found to be restricted to root cells, particularly to endodermis and cortex parenchyma cells [81]. In *Eupatorium cannabium* (Asteraceae, Eupatorieae) HSS expression was also found in the cortex parenchyma cells but not in the endodermis. In addition, HSS expression was shut down when the flower buds opened [82]. In contrast, in *Phalaenopsis* (Orchidaceae) HSS is expressed in the tips of aerial roots and in young flower buds [83]. In the Boraginaceae different HSS expression patterns were observed. In *Heliotropium indicum*, HSS was expressed exclusively in non-specialized cells of the lower epidermis of young leaves and shoots while in *Symphytum officinale* HSS expression was detected in the cells of the root endodermis and in leaves underneath developing inflorescences. In *Cynoglossum officinale* HSS expression was only observed in roots. In young roots, its expression was limited to cells of the endodermis, while in later developmental stages cells of the pericycle, it also showed HSS expression [84].

One theory regarding establishment of secondary metabolic pathways suggests that changes in gene function lead to subfunctionalization and a subsequent duplication event to two genes with different but complementary subfunctions, preserving the original enzyme function [75]. This complements an early suggestion by [85] that, prior to a gene duplication leading to novel protein function, the original gene was bifunctional. This theory is also supported by the bifunctionality of DHS, when considering this model for the explanation of HSS evolution. Furthermore, gene regulation and therefore gene expression patterns, might vary for the two genes, resulting from gene duplication [86]. This suggestion finds support in the varying expression patterns of DHS and HSS described above.

While extensive research on the recruitment of HSS has been conducted, there is still little knowledge about the evolution of the entire PA biosynthetic pathway, as it cannot be explained by the presence of homospermidine alone. Introduction of HSS into non-PA producing plants only results in formation of homospermidine, rather than of PAs or any precursors downstream of homospermidine [87].

#### 3.1.2. Copper-Dependent Diamine Oxidases and Cyclization of the Dialdehyde

Early evidence for involvement of a diamine oxidase in the incorporation of homospermidine into necine bases was provided by incubating homospermidine in vitro with a diamine oxidase fraction prepared from pea. Reduction of the reaction products by sodium borohydride and subsequent analysis by GC revealed that mainly trachelanthamidine and a small amount (approximately 5%) of isoretronecanol were obtained. In another experiment, reduction of the reaction products was performed with liver alcohol dehydrogenase, which again mainly yielded trachelanthamidine and little isoretronecanol [88]. However, the stereochemical properties of the obtained products were not investigated. Subsequent studies confirmed these initial findings and suggested involvement of a copper-dependent diamine oxidase since treatment of *Senecio vulgaris* and *Heliotropium indicum* with 2-hydroxyethylhydrazine (HEH), a potent diamine oxidase inhibitor, caused homospermidine accumulation and impeded PA biosynthesis [71,89].

Oxidation of amine precursors followed by cyclisation is a common theme in alkaloid biosynthesis. In tropane biosynthesis *N*-methylputrescine is oxidized by methylputrescine oxidase to 4-methylaminobutanal, which cyclizes spontaneously to the *N*-methyl-Δ^1^-pyrrolinium cation (Figure 9) [9]. This is similar to the first steps proposed for conversion of the dialdehyde 4,4′-iminodibutanal to the pyrrolium cation (Figure 10). However, biosynthesis of necine bases continues with reaction of the remaining aldehyde group for closure of the second ring via a Mannich-type reaction mechanism, which leads ultimately to pyrrolizidine-1-carbaldehyde. While cyclization in tropane alkaloid biosynthesis might be spontaneous, this is rather unlikely for necine base synthesis. Spontaneous cyclization of the dialdehyde and subsequent reduction would result in a mixture of the four saturated 1-hydroxymethylpyrrolizidines (±)-trachelanthamidine and (±)-isoretronecanol. After desaturation, a mixture of (±)-supinidine would be obtained and subsequent C-7 hydroxylation would, depending on the specificity of the hydroxylase, lead to a mixture of (+)-retronecine and (-)-heliotridine or (-)-retronecine and (+)-heliotridine. However, such mixtures are usually not observed in PA-producing plants. In contrast, plants usually contain PAs with necine bases of only a specific stereochemical configuration. For instance, *Senecio jacobaea* and *Senecio aquaticus* contain, apart from otonecine-type PAs, only senecionine-like, jacobine-like and erucifoline-like PAs, which are all (+)-retronecine-type PAs [90]. *Crassoceopahlum crepidioides* contains only jacobine, a PA of the (+)-retronecine type [50]. *Lolium perenne* contains only the (-)-isoretronecanol type PAs *Z*- and *E*-thesinine and its rhamnosides [91]. *Borago officinalis* contains only alkaloids of the (-)-isoretronecanol, (-)-supinine and (+)-retronecine-type [92], which have the same stereoconfiguration of the C-8 hydrogen. *Heliotropium europaeum* contains only (+)-heliotridine-type PAs [93]. The observed stereochemical specificity argues clearly against spontaneous cyclisation and suggests an enzymatic mechanism. Since enzyme-catalyzed cyclization must immediately follow oxidative deamination it is tempting to speculate that the copper-dependent diamine oxidase might also either support stereospecific cyclization of the dialdehyde or act in a protein complex with a second enzyme that catalyzes cyclization. 

#### 3.1.3. Further Downstream Reactions

After cyclization the formed pyrrolizidine-1-carbaldehyde is reduced to the alcohol. This might be catalyzed by an alcohol dehydrogenase (ADH), since Robins showed that ADH can in principle reduce the carbaldehyde to the alcohol [88]. Additional evidence comes from a detailed stereochemical study of necine base formation. Feeding of *Senecio isatideus* with deuterium labeled [1,1,4,4-^2^H_4_]-putrescine and hydrolysis of the obtained PA yielded retronecine that retained three deuterium atoms on the right handed ring: Two at C-3 and one at C-9. Importantly, the latter was in the *S* position, which is the stereochemistry expected for an ADH-catalyzed reduction of an carbaldehyde [59].

The sequence of retronecine base interconversions was studied by Birecka and Catalfamo, using pulse-chase experiments [94]. *Heliotropium spathulatum* was used for this study since this species produces (-)-trachelanthamidine, (-)-supinidine and (-)-retronecine containing PAs. The plants were treated with [^14^C]-carbon dioxide for 2 h prior quenching incorporation with unlabeled carbon dioxide. Samples were taken after 12 h, 24 h and 48 h and the specific activity of the necine bases analyzed. The activity of (-)-trachelanthamidine increased first, which was followed by an increase of the activity of (-)-supinidine and finally (-)-retronecine. This suggests that (-)-trachelanthamidine is first dehydrogenated at the C-1/C-2 bond to (-)-supinidine, which is subsequently hydroxylated at C-7 to (-)-retronecine. However, the involved enzymes remain elusive in addition to whether the free necine bases are modified or the, at least partially, esterified PAs. 

### 3.2. Biosynthesis of Necic Acids

While necine bases are synthesized by a common pathway, different necic acids are formed by distinct modes. A number of acids found in PAs are normally present in plants. This includes, for instance, acetic acid, benzoic acid, and *p*-coumaric acid. These acids and their activated forms, the coenzyme A thioesters, are formed by common metabolic pathways of primary metabolism, which will not be discussed here. 

#### 3.2.1. Tiglic Acid and Related C_5_ Necic Acids

C_5_-acids of the tiglic acid type are frequently observed as building blocks of secondary plant metabolites. For instance, tiglic acid is the moiety of meteloidine and other tropane alkaloids formed by *Datura* species [95]. Feeding experiments in *Datura meteloides* showed that tiglic acid is derived from isoleucine [96]. Similar results were also obtained in *Cynoglossum officinale* for angelic acid, which is present as ester in the PA heliosupine [31]. Feeding of *Datura meteloides* with radiolabeled 2-methylbutanoic acid showed that the radioactivity was efficiently incorporated into the tiglic acid moiety of meteloidine, identifying 2-methylbutanoic acid as a precursor [97]. The same pathway was also established for carabid beetle [98]. Data from mammals [99] suggest that 3-hydroxy-2-methylbutyric acid acts as intermediate between 2-methylbutanoic acid and tiglic acid, and that the intermediates appear as coenzyme A thioesters. These steps are in fact part of the common l-isoleucine degradation pathway and thus it is not surprising that these metabolites including tiglyl-CoA are present in most plant tissues. McGaw and Woolley showed that angelic acid is derived by *cis*-*trans* isomerization from tiglic acid in *Cynoglossum officinale* [100]. These data suggest a pathway for biosynthesis of tiglic and angelic acid starting from l-isoleucine (Figure 11). 

#### 3.2.2. C_7_ Necic Acids

The more complex acids are generated by specific biosynthetic pathways. The observation that many of these acids include one or two C_5_ units resembles isoprenoids, although with unusual linkage and oxygenation patterns, encouraging the idea that they might be derived from mevalonate [101]. However, feeding experiments with ^14^C-labeled acetate, acetoacetate and mevalonate did not result in specific incorporation into necic acids, excluding these compounds as direct precursors for necic acids. In contrast, feeding experiments with ^14^C-labeled amino acids provided compelling evidence that necic acids are derived from l-isoleucine and l-valine. Additionally, [^14^C]-l-threonine and a few further amino acids were found to be efficiently incorporated, but only since they acted as precursors for l-isoleucine or l-valine.

The PA heliosupine appearing in *Cynoglossum officinale* consists of the necine base (-)-heliotridine esterified on the 7-hydroxy group with angelic acid (which is synthesized as shown in Figure 12) and on the 9-hydroxy group with echimidinic acid, a C_7_-acid (Figure 12A). Feeding experiments revealed that echimidinic acid is derived from l-valine and an additional C_2_ unit of unknown origin [102].

Trachelanthic acid, which is structurally very similar to echimidinic acid, is also derived from l-valine, as revealed by feeding of ^14^C-labeled amino acids. These results were further supported by an isotopologue study by feeding a root culture of *Eupatorium clematideum* with a mixture of ^13^C-labeled and unlabeled glucose [103]. Subsequently, trachelanthamine and the amino acids l-arginine, l-proline, and l-valine were isolated and the ^13^C labelling pattern analyzed by NMR spectroscopy. The ^13^C pattern of the necine base was in agreement with that of l-arginine, while the major part of trachelanthic acid (shown in red in Figure 12B) fitted to that observed for l-valine. The remaining two carbons showed a pattern as reconstituted for hydroxyethyl-TPP. This confirmed the conclusions for echimidinic acid. However, it must be emphasized that the result that the ^13^C pattern of the C_2_ unit corresponds to that of hydroxyethyl-TPP does not mean that this compound is indeed the direct precursor. In contrast, there might be a number of metabolites in-between. Thus, the direct precursor of the C_2_ unit remains to be determined. 

#### 3.2.3. Monocrotalic Acid and Related Compounds

Biosynthesis of monocrotalic acid was analyzed in the Fabaceae *Crotalaria retusa* and *Crotalaria spectabilis* by feeding of ^14^C-labeled acetate, DL-alanine, l-threonine, and l-isoleucine [104]. While minute incorporation was observed for acetate and DL-alanine, l-threonine and L-isoleucine were incorporated into monocrotaline to significant levels. Degradation studies showed that the necine base had incorporated little radioactivity while the necic acid monocrotalic acid contained most of the activity. More specifically, most activity was observed in fragments consisting of C-1, C-2 and C-6 and of C-3 and C-7, while the fragment consisting of C-4, C-5, and C-8 had incorporated comparatively little activity (Figure 13A). Thus, the right hand part of monocrotalic acid is derived from l-isoleucine while the origin of the left hand part remains elusive. Another study found, in addition to efficient incorporation of ^14^C-labeled l-isoleucine and l-threonine, a significant incorporation of l-valine [105]. Since no degradation studies were performed, it remains unknown whether l-valine serves as a precursor for the left hand part.

The structure of trichodesmic acid is very similar to monocrotalic acid, except that a methyl group is replaced by an isopropyl group (Figure 13B). Thus, it was also expected that the right hand part of trichodesmic acid is derived from l-leucine (and l-threonine), which could be confirmed by feeding studies. In addition, ^14^C-labeled l-valine and l-leucine were also efficiently incorporated, particularly into the right hand part [106]. The fact that both amino acids were incorporated to a similar level suggests that they are converted to a common precursor prior to incorporation into trichodesmic acid. However, the pathway and mechanism remain to be elucidated. 

#### 3.2.4. Senecic Acid and Senecic Acid-Derived Compounds

Among the necic acids forming 12-membered macrocyclic PAs biosynthesis of senecic acid is best characterized. Since seneciphyllic acid and isatinecic acid are derived from senecic acid, results obtained for these compounds will be discussed together with that for senecic acid. Feeding experiments of *Senecio douglasii* with ^14^C-labeled acetate, acetoacetate and mevalonate showed that these compounds were inefficiently incorporated into seneciphylline, and that the radioactivity was randomly distributed between the necine base retronecine and the necic acid seneciphyllic acid. This indicated that none of the tested compounds are direct precursors of the alkaloid. In contrast, l-threonine was efficiently incorporated and the radioactivity was found selectively in the necic acid part. Subsequent experiments showed that l-threonine was first converted to l-isoleucine prior to incorporation into seneciphyllic acid, identifying isoleucine as the direct precursor. In addition, evidence was provided that only C-2 to C-5 of l-leucine were incorporated into seneciphyllic acid, while C-1 (the carboxy group) was lost [107]. Feeding experiments using *Senecio magnificus* provided similar results for senecic acid and provided compelling evidence that the acid is derived from two molecules of l-isoleucine, both losing their C-1 during biosynthesis [108,109]. From the four possible isoleucine stereomers, l/d-isoleucine and l/d-alloisoleucine, only l-isoleucine was efficiently incorporated [110]. The loss of the carboxy groups raised the question for the five-carbon intermediate. Possible candidates were 2-methylbutanoic acid, which is an intermediate of l-isoleucine degradation and thus are present in most plant tissues, and 2-methyl-3-oxobutanoic acid and angelic acid. The latter was selected since it has the same configuration as the double bond in senecioic acid. Feeding resulted in incorporation of 0.06% (*S*)-[^14^C]-2-methylbutanoate, which was a slightly lower level than for l-[^14^C]-leucine, where incorporation rates of 0.1 to 0.4% were observed. However, analysis of the necic acid and the necine base showed random distribution of radioactivity. The two other compounds were incorporated in only trace amounts. Based on these results the idea that one of those compounds is a precursor of senecic acid was discarded. Next, ^14^C-labeled 2-amino-3-methylenepentanoic acid (β-methylenenorvaline) was tested and efficient incorporation (0.07–0.11%) was observed with essentially only senecioic acid possessing the radio label. This observation suggested this compound as a possible intermediate (Figure 14A). Unfortunately, it could not be investigated whether the ^14^C-label was incorporated in either or both halves of senecic acid [111]. 

The most enigmatic step in biosynthesis of senecic acid and related compounds is the mechanism for uniting C-13 and C-14 (Figure 14A). C-13 corresponds to C-4 carbons of the isoleucine precursor forming the right hand part of senecic acid. Studies with l-leucine stereospecifically labeled at C-4 with ^3^H showed that the pro*-R* hydrogen was retained, while the pro*-S* hydrogen was lost on both l-isoleucine molecules required for synthesis of senecic acid. This demonstrates that both carbons cannot be oxidized further than the alkene or carbinol level [113]. The observation that 2-aminobutanoic acid is efficiently converted to isoleucine in feeding experiments [113] (Figure 14C) enabled feeding studies with stable isotope labelling and analysis of the produced alkaloids by NMR [112]. Feeding of *Senecio pleistocephalus* and root cultures of *Senecio vulgaris* with [3,4-^13^C_2_]-2-aminobutanoic acid confirmed that C-3 and C-4 of 2-aminobutyric acid carbons are the precursors of C-20 and C-21 in the left hand side of senecic acid and of C-13 and C-19 in the right hand side (Figure 14D). Analysis of alkaloids obtained by feeding of [3,4-^2^H_5_]-2-aminobutanoic acid revealed that three ^2^H were retained on the methyl groups of both C-21 and C-19, and one ^2^H was retained on each C-13 and C-20. These results show that the C-3 and C-4 positions of 2-aminobutanoic acid corresponding to C-5 and C-6 of l-isoleucine are exclusively and equally incorporated intact into the two halves of senecic acid. These results are in agreement with a mechanism proposed by Bale, where the left hand part of senecic acid is derived from l-isoleucine by conversion to β-methylenenorvaline, which reacts with pyridoxal phosphate to a Schiff’s base (Figure 14E). Deprotonation would generate a mesomeric anion with a nucleophilic character at C-6 (numbering corresponding to that of l-isoleucine). This nucleophile might react with a suitable, yet unknown electrophile [111]. From the experiments with ^3^H-labeled l-isoleucine and ^2^H-labeled 2-aminobutanoic acid, it is clear that the electrophile cannot possess a carbonyl group at C-4 while an alkene would be in agreement with those results. An attractive candidate is tiglyl-coenzyme A since this compound is a metabolite of l-isoleucine degradation [114] and thus present in most tissues. In addition, the C-C double bond is activated due to the neighboring keto group. Attack of the nucleophile on tiglyl-CoA would create the C-13/C-14 bond. Subsequent release of the pyridoxal phosphate coenzyme would create the C-16/C-20 double bond, which is either directly or further metabolized (e.g., as epoxide, diol, etc.) almost invariably present in necic acids of the senecic acid type. It must be emphasized that feeding experiments using (*S*)-[^14^C]-2-methylbutanoate, the precursor of tiglyl-CoA in the l-isoleucine degradation pathway, did not give conclusive results since high incorporation was observed, but the incorporation pattern was unspecific [111]. However, it might be worth reinvestigating a possible role of tiglyl-CoA in senecic acid biosynthesis by state of the art isotopologous NMR-based techniques.

First evidence for conversion of senecionine to other senecionine-type PAs came from feeding of [^14^C]-putrescine to *Senecio vernalis* root cultures, which resulted in rapid incorporation in senecionine *N*-oxide. However, the labeled senecionine *N*-oxide was progressively converted within 10 days to senkirkine, an otonecine-type PA [28]. Pulse-chase feeding experiments of *Senecio erucifolius* root cultures with [^14^C]-putrescine showed rapid incorporation of radioactivity in senecionine *N*-oxide and revealed absence of any significant alkaloid turnover with the exception of a slow but progressive conversion of labeled senecionine *N*-oxide to its dehydrogenation product, seneciphylline *N*-oxide [115]. Using whole *Senecio erucifolius* plants, in addition to formation of seneciphylline *N*-oxide, conversion of senecionine *N*-oxide to *O*-acetyl seneciphylline, and of erucifoline to *O*-acetyl erucifoline and to eruciflorine was also observed, indicating that the stem might be crucial for these modifications [116]. These data clearly demonstrate that the first product of PA biosynthesis in *Senecio* is senecionine *N*-oxide, which is further converted to a bouquet of senecionine-like alkaloids or their *N*-oxides (Figure 15). Such reactions mainly include simple one or two-step reactions like hydroxylations, acetylations, desaturations and epoxidations. In addition, it is tempting to speculate that the epoxide ring of jacobine and similar compounds might undergo further metabolization by hydrolysis or addition of hydrochloric acid to yield jacoline *N*-oxide or jaconine *N*-oxide, respectively. Thus, senecionine-like alkaloids are derived by modification of senecionine *N*-oxide rather than combination of necine bases with different necic acids.

## 4. Regulation of Pyrrolizidine Alkaloid Levels and Biosynthesis

PA biosynthesis is known to be regulated differently during plant development. For example, in *Eupatorium cannabium* HSS expression is restricted to young roots with close correlation to plant growth [82]. HSS is only detectable in newly grown, white roots until the produced biomass peaks and flowers open. When these plant parts die off at the end of the vegetation period, de novo alkaloid biosynthesis is required at the beginning of the next growth period. In contrast, plants of the species *Symphytum officinale* can activate a second site of HSS expression once inflorescence development begins in leaves subtending emerging flowers [117]. The same study could also demonstrate that not only HSS but the whole PA biosynthetic pathway are active in these young leaves. This second site of PA biosynthesis allows for drastically increased PA levels within the inflorescences ensuring optimal protection of the reproductive structures against herbivores.

Furthermore, PA expression is influenced by nutrient and water supply as well as herbivore infestation [118,119,120]. On the effects of nutrient availability, a general proposal is that a higher NPK supply reduces the PA content in *Senecio jacobaea*, *Senecio vulgaris,* and *Senecio aquaticus*, presumably due to dilution effects in shoots and roots, whereas PA levels in flowers are not affected [118]. These findings are further supported by results of PA analysis in *Senecio jacobaea* and *Senecio aquaticus* on marginal, sandy soils showing that low nutrient supply increases relative PA amounts [119]. Nevertheless, the same study found a significant change in PA composition on fertilized soils, as they reported a high rise in jacobine contents, while the total PA amount remained constant under nutrient rich conditions. One reason for increased jacobine levels could be the role of jacobine in insect herbivore resistance. Research on root herbivore damage in *Senecio jacobaea* has revealed no change in total PA content in the whole plant, but a translocation of PAs from shoot to the root, mostly of *N*-oxides, also causing the ratio of *N*-oxide to free PAs to drop from 2:1 to 1:1 in the shoot [120]. Moreover, plants stabilize free PA levels in aerial plant parts by converting PA *N*-oxide into free bases [120]. 

There is substantial evidence for the genetic control and heritability of PA biosynthesis. A study by Vrieling et al. [121] using a diallel cross of *Senecio jacobaea* revealed that 48% of the variation in total PA content was caused by genotypic variation. Half-sib analysis from natural progenies combined with drought and nutrient deficiency treatments indicated that 85–90% of the differences in total PA concentration were caused by additive genetic variation. Heritability for individual PAs was significant as well. van Dam et al. [122] estimated that 33–43% of the variation in PA levels between selfed families of *Cynoglossum officinale* is due to broad-sense heritability and found that the inducibility of PA production by wounding of leaves differs significantly between families. Macel et al. [123] evaluated the variations in PA profiles within clonal families of *Senecio jacobaea* and found significant differences in the relative percentages and absolute concentrations of individual PAs between clonal families. The variation in PA composition within clonal families was smaller than the variation among families, indicating a strong genetic influence. Furthermore, Joosten et al. [51] found that the presence of pyrrolizidine alkaloids in the tertiary amine form in *Senecio jacobaea* is genotype-dependent as well. Pelser et al. [124] reconstructed the evolutionary history of qualitative PA variation in the *Jacobaea* section of *Senecio*. They concluded that the variations in PA profiles are not caused by simple gain and loss of PA-specific genes, but rather by changes in transient expression of PA biosynthesis genes, since the large intra and interspecific variation in PA distribution seems to be largely incidental and nearly all of the PAs identified in the *Jacobaea* section are also present in species of other sections of the genus.

Hybridization can result in the occurrence of novel PA structures through inter-specific epistatic interactions between enzymes and substrates, as demonstrated by Kirk et al. [119]. They observed that F1 hybrids between *Senecio jacobaea* and *Senecio aquaticus* produce florosenine, which is not present in parent populations. Since florosenine is formed by *O*-acetylation of otosenine, the capacity of *Senecio aquaticus* to synthesise otosenine and the ability of *Senecio jacobaea* for PA acetylation could be combined in the hybrids. A study by Cheng et al. [125] using F1 and F2 hybrids of *Senecio jacobaea* and *Senecio aquaticus* showed that hybrid roots contained acetylated otosenine-like PAs, while roots of the parental lines did not. Additionally, shoots of F2 hybrids exhibited an over-expression of otosenine-like PAs with contents reaching >20% of total PAs. In contrast, otosenine-like PAs have not been previously reported as a major fraction of the PA bouquet in *Senecio jacobaea* or *Senecio aquaticus*. Furthermore, some F2 hybrids contained higher relative proportions of erucifoline-like PAs in shoots compared to the parental genotypes where jacobine or senecionine-type PAs dominated. These findings indicate that hybridization contributes to the increase of the structural diversity of PAs. 

There is little knowledge about the regulation of PA biosynthesis by plant hormones. Methyl-jasmonate is thought to play a part because of its role as an elicitor of induced responses and anti-herbivory resistance. Wei et al. [126] conducted a study with *Senecio jacobaea* and *Senecio aquaticus* grown aseptically on medium containing methyl jasmonate. In treated *Senecio jacobaea* plants, the total concentration of PAs increased in shoots but decreased in roots. A similar non-significant trend was observed for *Senecio aquaticus*. The application of methyl jasmonate leads to a strong shift from senecionine to erucifoline-like PAs, while the jacobine and otosenine-like PAs were not affected. This indicates that methyl jasmonate does not necessarily induce *de novo* synthesis, but rather leads to reallocation of certain PAs from roots to shoots and a conversion of PA structures. Sievert et al. [127] tested the influence of methyl jasmonate on PA levels and on the transcript levels of homospermidine synthase in *Heliotropium indicum*, *Symphytum officinale*, and *Cynoglossum officinale,* but could not detect any significant effects. The only case where a clear influence of methyl jasmonate on PAs was observed was in hairy root cultures of *Echium rauwolfii* [128]. Root culture medium, supplemented with 100 μM of methyl jasmonate, lead to a 19-fold increase of total PAs, while the flavonoid quercetin boosted the PA accumulation 6-fold at 50 μM. When the root cultures were pre-incubated with salicylic acid, the inducing effect of both compounds could be suppressed.

Regarding PA transport within the plant it has been shown for *Senecio vernalis* that translocation from roots to shoots occurs through phloem cells, located opposite to specialized HSS-expressing cells of the endodermis and the adjoining cortex parenchyma [81,129]. These cells are likely to execute the entire biosynthesis of PAs which can then be transported as *N*-oxides through the pericycle into the phloem, from there into the shoot and finally to the sites of storage, thereby uncoupling sites of PA synthesis from sites of activity. PA *N*-oxides are more soluble than free PA bases and are thus very phloem mobile, enabling transport between different storage tissues when necessary [130]. Also, the shoot is suggested to harbor the ability to diversify the chemical composition of PAs by simple hydroxylation, epoxidation, dehydrogenation, or *O*-acetylation reactions [89,116].

In a study by van Dam et al. [131], the inducibility of PA production after leaf damage was tested in two different plants species, namely *Senecio jacobaea* and *Cynoglossum officinale*, by cutting off 50% of their leaf area. PA concentrations were measured at different time points after damage and the cut-off leaf tips were used as controls for diurnal fluctuations. In *Senecio jacobaea* leaf damage significantly decreased PA concentrations. PA levels reached a minimum 12 h after the treatment and went back to initial values 24 h after. PA levels in *Cynoglossum officinale* steadily increased with time after damage. The authors hypothesize that the different responses might be due to adaptations related to the type and severity of herbivory occurring under natural conditions as *Senecio jacobaea* is adapted to initiate regrowth after severe defoliation by specialist herbivores. Therefore, a decrease of PAs in the leaves after damage could be caused by a reallocation of defense resources for future regrowth. *Cynoglossum officinale* on the other hand is not adapted to severe herbivory and does not recover as quickly following damage. Thus, it might be advantageous for this species to boost its chemical defense in order to reduce herbivore-inflicted damage.

## 5. Biological Activity

### 5.1. Role in Plant Ecology

The parallel evolvement of PA synthesis in phylogenetically unrelated plant families suggests an evolutionary advantage from PA presence. One important aspect in this context is, as for the evolution of many other secondary metabolites, a defense against insect herbivores [132,133]. Support for this hypothesis is found by studying insects specialized on PA-containing plants, which are found in a number of families, such as Lepidoptera, Coleoptera, Orthoptera, and Homoptera [134]. Additionally, grazing animals are known to only consume PA plants in times of low feed supply and generalist insect herbivores are suggested to be deterred by different PAs whereas adapted insects are thought to be attracted [131,134,135,136]. The general PA metabolism in insects is similar to human PA metabolism, including the negative physiological effects. Various studies regarding the effects of single PAs and general PA content on both specialist and generalist herbivore feeding have been conducted. Wei et al. [137] and Macel et al. [138] showed that the influence on different generalist species depends on PA-type and concentration. While Macel et al. [138] only observed effects under certain dietary conditions, Wei et al. [137] performed bioassays with *Senecio jacobaea* x *Senecio aquaticus* F2 hybrids with varying PA contents to draw conclusions about the correlation between the presence and abundance of different PAs and feeding damage. The results indicate that, for instance, thrips prefer leaves with lower jacobine-like PA contents, while for slugs, low senecionine-like PA levels are important. Furthermore, single PAs of both types were correlated to feeding behavior. These results are particularly interesting because presence of a C-13/C-19 double bond seems to play an important role in generalist herbivore resistance [137]. However, also jacobine *N*-oxide, although not carrying the C-13/C-19 double bond, has been shown to deter western flower thrips (*Frankliniella occidentalis*) and higher amounts were shown to be a characteristic trait of thrips resistant PA plants [90,139].

There are several reports stating that PA *N*-oxides are less bioactive against insect herbivores than the corresponding free bases [131,140,141,142]. Contrarily, PAs are mostly present in plants as *N*-oxides [129] with some jacobine-like PAs occurring up to 50% as free base in *Senecio jacobaea* [51] and *Crassocephalum crepidioides* [50]. An advantage of *N*-oxides is their higher solubility resulting in more efficient storage and transport [130,143,144]. Another possible explanation suggested by Liu et al. [145] are synergistic effects of PA *N*-oxides with other plant metabolites. They showed that PA free bases and chlorogenic acid act antagonistically on western flower thrips (*Frankliniella occidentalis*) mortality, while in contrast, PA *N*-oxides showed synergistic interactions with chlorogenic acid on thrips mortality. In the absence of chlorogenic acid, PA free bases decreased thrips survival more severely than PA *N*-oxides, but when chlorogenic acid was added, this effect was reversed. Thus, the bioactivity of individual PAs seems to be influenced by the natural chemical background in which they occur. This aspect is further supported by a study of Liu et al. [146], which demonstrated that fractions of a methanol extract from *Senecio jacobaea* all showed a higher survival rate of western flower thrips than the whole extract. Additionally, the expected combined effect of the single fractions on survival, assuming no interaction, was lower than that of the methanol extract. Furthermore, retrorsine and retrorsine *N*-oxide were added alone and in combination to the five fractions; the effects on thrips survival depended on the fraction to which the PAs were added. These studies highlight the relevance of synergistic effects of PAs and other plant metabolites on herbivores. In contrast, different PAs (free amines and *N*-oxides) do not show synergistic effects on one another [141].

While the correlation of PA content and feeding is generally negative for generalists, the exact opposite is reported for specialists, indicating an attraction by PA in plants. In fact, several cases of adaption and utilization of PAs by insects have been reported [1,147,148]. The oviposition of *Tyria jacobaea*, a well-studied member of the Lepidoptera, also called the cinnabar moth, was shown to be positively influenced by the concentration of jacobine-like PAs [136]. This finding suggests an advantage for *Tyria jacobaea* larvae from PA ingestion. Frequently, PAs are sequestered and stored as *N*-oxides in specialist beetles, moths, butterflies and grasshoppers [1,147,148,149]. With the storage of PA *N*-oxides insects have developed a strategy of using plant defense chemicals for their own defense against predators. An astonishing example for this adaption are neotropic Ithomiinae butterflies which, unlike moths, neither feed on PA plants nor sequester them from larvae stage on, but rather take up PAs solely through nectar or withered twigs [135,138]. This habit protects them from the spider *Nephila clavipes*, which cuts out butterflies that had previously ingested PAs, of her own net [150,151]. In *Utetheisa ornatrix* the male butterflies are able to transfer their PA storage during mating onto females, who then utilize them to protect their eggs [152]. Sequestered PAs play an important role in the mating process of some tribes of the Lepidoptera family, because they serve as a precursor for male pheromones [135,153]. Another impressive adaption was found in *Tyria jacobaea* and *Creatonotos transiens*. These two arctiids are able to synthesize their own PAs by esterifying a necine base of plant origin with a necic acid derived from isoleucine by their own metabolism [42,154,155]. Kubitza et al. [156] provided a high-resolution crystal structure of the flavin-dependent monooxygenase from the African locust (*Zonocerus variegatus*). This locust expresses three flavin-dependent monooxygenase isoforms contributing to a counterstrategy against PAs. By *N*-oxidation of PAs and accumulation of PA *N*-oxides within its hemolymph the locust circumvents the chemical plant defense and uses PA *N*-oxides to protect itself against predators. By such mechanisms specialized insects may gain advantages from PA-containing plants. Macel and Klinkhamer [157] reported positive correlations between the jacobine *N*-oxide and free base content of *Senecio jacobaea* with damage from specialist insect herbivores. Moreover, a study by Joshi and Vrieling [158] reported higher jacobine concentrations in *Senecio jacobaea* growing in invasive areas, indicating fitness advantage of lower PA amounts in areas with specialist herbivores compared to areas with exclusively generalist insects, wherein higher contents are advantageous.

Livshultz et al. [77] studied the evolution of PAs in species of the Apocynaceae family, which are larval host plants for PA-adapted butterflies of the Danainae family. The phylogenetic analysis showed a monophyletic origin of the HSS sequences early in the evolution of one Apocynaceae lineage. They found HSS orthologues, pseudogenes and multiple losses of HSS amino acid motifs in several non-PA producing species consistent with multiple independent losses of PAs. This indicates a selection for the loss of PA biosynthesis by PA-adapted specialist herbivores in the Apocynaceae family.

Despite their contribution in defense against insects, PAs have also been reported to influence interactions with symbiotic and pathogenic fungi. An inhibitory effect of PAs was measurable on *Fusarium* and *Trichoderma*, with PA mixtures exhibiting the highest inhibition [159]. Interestingly, in the same study an enhancement of growth by PAs was quantifiable for strains isolated from *Senecio jacobaea* indicating specialization. PA chemotypes of *Senecio jacobaea* also influence the diversity of fungal communities in soil, as shown by Kowalchuk et al. [119]. It was possible to distinguish fungal communities associated with high-PA jacobine chemotypes from low PA samples as well as senecionine/seneciphylline chemotypes. Additionally, a trend towards lower diversity in the rhizosphere of high-PA plants was observed compared to low-PA plants. This shows that PA chemotypes of *Senecio jacobaea* influence fungal communities in the rhizosphere, with jacobine types selecting more intensely than senecionine/seneciphylline types. In another study, artificial root colonization of *Senecio jacobaea* by *Rhizophagus irregularis* increased concentrations of senecionine, jacoline *N*-oxide, jaconine *N*-oxide, and usaramine *N*-oxide in roots but not in shoots. Only the amount of senecionine was significantly correlated with root length colonized [160]. On the other hand, Reidinger et al. [161] observed a negative correlation between natural colonization levels of roots of *Senecio jacobaea* by vesicles and the concentrations of both jacoline and total PAs. The authors suggest that the natural variations in PA concentrations between individual plants might have affected arbuscular mycorrhizal fungi colonization.

In the Fabaceae member *Crotalaria* PA production can also be influenced by root nodulation as demonstrated by Irmer et al. [162]. Only nodulating *Crotalaria spectabilis* plants infected with their rhizobial partner produce the PA monocrotaline, which is not regarded as being functionally involved in the symbiosis. The absolute amounts of PA per plant were highest in leaves, followed by nodules, roots, and stems, while the concentration was highest in the nodules (1.97 mg/g dry weight), exhibiting a 10-fold higher concentration than in leaves (0.21 mg/g dry weight). A plant derived HSS sequence was identified suggesting that the plant and not the microbiont is the PA producer. HSS transcripts were only detectable in nodules, indicating that they are the only location of alkaloid biosynthesis and the source from which the PAs are transported to above ground parts of the plant.

Plants which are not producing PAs can still accumulate them if they grow on soil containing decomposing PA-containing plants according to a study by Nowak et al. [163]. Various plant species commonly used as herbal teas or spices, i.e., melissa (*Melissa officinalis*), peppermint (*Mentha x piperita*), chamomile (*Matricaria chamomilla*), and parsley (*Petroselinum crispum*), were grown on soil mulched with 1 g of dried *Senecio jacobaea* plant material to investigate the uptake of PAs. Seven days after application, all mulched plants exhibited marked concentrations of PA in their leaves while the untreated controls were PA-free. The maximum PA levels in peppermint, melissa and chamomile were 0.1–0.15 mg/kg dry weight, whereas PA levels in parsley were up to five times higher (>0.5 mg/kg). Fourteen days after mulching, PA concentrations severely decreased, especially the *N*-oxide forms. The composition of the imported PAs in parsley, chamomile and melissa, showed a similar pattern to that of *Senecio jacobaea* with erucifoline being the most abundant alkaloid, followed by seneciphylline and jacobine, whereas in peppermint erucifoline occurred only in traces. This observation is probably due to species dependent differences in PA degradation or modification in the acceptor plants. While PA free bases can pass the membranes of the roots by simple diffusion, the uptake of PA *N*-oxides might be catalyzed by transporters normally responsible for the uptake of other compounds. Since the flowers of treated chamomile plants did not contain any PAs it was assumed that the transport to the leaves of acceptor plants happens via the xylem. These findings are significant for the production of herbal teas, spices, and plant derived pharmaceuticals as PA-contaminations can occur due to decomposing PA-containing plants in the fields.

### 5.2. Toxicity of Pyrrolizidine Alkaloids and Mechanisms for their Detoxification

Toxicity and pharmacology of PAs has recently been discussed in a detailed review by Moreira et al. [10]. Thus, we summarize toxicity here only very briefly and focus more on detoxification mechanisms.

PAs are one of the most important classes of naturally occurring toxins due to their wide distribution and high risk of unwitting consumption of contaminated natural products like grains, honey, milk, herbal teas, and medicines [3,164,165,166,167,168,169,170,171]. Numerous studies have demonstrated the hepatotoxic [172], genotoxic [173], cytotoxic [173,174], tumorigenic [175], and neurotoxic [176] potential of naturally occurring PAs. Possible symptoms of PA toxicity are hepatic veno-occlusive disease, liver cirrhosis, megalocystosis and cancer. Furthermore, they can cause chronic pulmonary arterial hypertension and congenital anomalies [3,177]. Several cases of poisonings and poisoning outbreaks caused by PA contaminated food have been documented [3,178]. The two main sources of intoxication for humans are the consumption of cereal grain contaminated with seeds from PA-containing weeds and the use of alkaloid-forming herbs or herbal remedies for medicinal and dietary purposes [17,177]. 

The degree of harm caused by a specific PA relies on its necine base structure, since toxic PAs, like the retronecine, otonecine and heliotridine-types, generally carry a 1,2-unsaturated necine base. The first step of metabolic activation of PAs after ingestion and absorption is dehydrogenation catalyzed by cytochrome P450 monooxygenases (CYP3A4 and CYP2C19) of the liver (Figure 16) [179]. In retronecine and heliotridine-type PAs hydroxylation of the necine base occurs at the C-3 and C-8 position to form 3- or 8-hydroxynecine derivatives followed by spontaneous dehydration [180,181,182]. For otonecine-type PAs, oxidative *N*-demethylation of the necine base occurs, followed by ring closure and dehydration [183,184]. Those reactions lead to the formation of reactive dehydropyrrolizidine intermediates, also known as pyrroles or pyrrolic esters, followed by the formation of pyrrolizinium ions. The electrophilic metabolites are capable of binding nucleophilic -OH, -SH, or -NH functional groups on physiologically important macromolecules like glutathione (GSH), DNA and proteins [173,185,186,187,188,189]. Nucleophilic substitution of one or two glutathione molecules produces the GSH conjugates 7-, 7,9-di- and 9-glutathionyl-6,7-dihydro-1-hydroxymethyl-5*H*-pyrrolizine for excretion [173,183,190,191]. These GSH conjugates are considered to be detoxification metabolites but it has been reported that they also act as reactive metabolites, causing DNA adduct formation [172]. Additionally, glutathione *S*-transferases (GSTs) can mediate these reactions leading to species-dependent toxicity effects, as it has been observed that guinea pig hepatic GSTs catalyze GSH conjugation of the pyrrolic metabolites of jacobine whereas rat hepatic GST did not affect the reaction [192]. GSH conjugation is one of the most important detoxification reactions of PAs. If the levels of reactive intermediates are high enough, the cellular GSH pool can be depleted causing severe oxidative damage of liver tissues [193,194,195]. Dehydro-PAs, including 6,7-dihydro-7-hydroxy-1-hydroxymethyl-5*H*-pyrrolizine (DHP) [196,197], their C-7 and C-9 hydrolysis product, and the dehydration product (3*H*-pyrrolizin-7-yl)methanol [198] are all reactive alkylating species. They can bind to liver cellular proteins and DNA resulting in the formation of DNA and/or protein adducts as well as DNA-protein cross links, which are all associated with genotoxicity and carcinogenicity ultimately leading to veno-occlusive disease and/or tumors [175,199,200]. 

A study on the in vitro metabolism of lasiocarpine in liver microsomes of humans, pigs, rats, mice, rabbits, and sheep was compared and a tendency towards species-dependent toxicity was observed [201]. The highest levels of reactive metabolites like (3*H*-pyrrolizin-7-yl)methanol and GSH conjugates were detected in human microsomes, followed by pig, rat and mouse, all known to be susceptible to PA intoxication. The microsomes of rabbit and sheep, known to be more PA-resistant, exhibited lower metabolite levels. Interestingly, the scale of PA toxicity also depends on age and gender, since male rats are more prone to clavinorine and isoline intoxication than female rats [194,202,203] and young mice are more susceptible to retrorsine in comparison to adult mice [204].

One mode of detoxification catalyzed by flavin-containing monooxygenase (FMOs) is the *N*-oxidation of the necine bases of retronecine and heliotridine-type PAs to form PA *N*-oxides [205,206]. Otonecine-type PAs on the other hand cannot form PA *N*-oxides due to methylation of the necine base nitrogen. PA *N*-oxides can be conjugated for excretion and are therefore generally regarded as detoxification products, although they may also be converted back to their toxic parent PAs [181,207,208,209]. As it was demonstrated for senecionine, the susceptibility of an animal species to PA exposure depends on the rate of FMO-catalyzed conversion of the PA into its *N*-oxide leading to detoxification as well as the P450-catalyzed formation of DHP leading to toxicity [210]. In susceptible species such as rats, cattle, horses, and chickens, a high glutathione–DHP conjugation is observed. However, in sheep, a resistant species, low rates of glutathione conjugation combined with high rates of FMO-catalyzed conversion of senecionine into its *N*-oxide occur. 

Another mode of detoxification in mammal metabolism is the cleavage of the necine base-necine acid bond, mediated by unspecific esterases followed by phase II-conjugation and excretion. This process was shown to contribute to guinea pig resistance to PA toxicity and rat susceptibility, as guinea pigs have higher liver esterase activity in contrast to rats [211,212,213,214]. 

Recently, *N*-glucuronidation has been identified as a potential new metabolic pathway in humans and several animal species for the activation and detoxication of PAs [215]. 

PAs with a saturated necine base like platynecine-type PAs are known to be nontoxic. It has been demonstrated that in hepatocytes treated with platyphylline the metabolites being formed are mainly dehydroplatyphylline carboxylic acid and in lower amounts, platyphylline epoxide as well as platyphylline *N*-oxide [216]. Disparate in comparison to unsaturated PAs, which are activated to reactive pyrrolic esters causing the formation of hepatotoxic protein and/or DNA adducts, the saturated structure of the platynecine-type PAs causes the P450-mediated oxidative metabolism to generate nonreactive polar compounds that are water-soluble and easily excreted. 

### 5.3. Beneficial Properties of Certain PAs

Despite the toxic properties of 1,2-unsaturated PAs and their negative implications on human and animal health, some PAs have the potential to be used for the treatment of diseases and infections. One of the most consequential and positive characteristics of these PAs is their glycosidase inhibitory activity [217,218], potentially leading to antidiabetic, anticancer, fungicidal, and antiviral effects [219]. Glycosidases are a class of enzymes involved in selective hydrolysis of sugar bonds in glycoconjugates as well as the covalent attachment of carbohydrates to lipids, proteins, and peptides on the cell surfaces [220,221,222,223]. Disfunctional glycosidases can reduce the proliferation of malignant cells and tumor growth [219,224]. Furthermore, inhibited glycosidases can cause the formation of disorganized oligosaccharides, thereby interrupting cell–cell and cell–virus recognition [219,225]. Thus, certain PAs may be used for the treatment of cancer, diabetes, and infections [226,227]. Some of the most potent glycosidase inhibitors are polyhydroxylated PAs since they mimic the pyranosyl and furanosyl core of the natural glycosidase substrates [226,228]. It has been demonstrated that the polyhydroxylated PAs alexine from *Alexa leiopetala* and australine from *Castanospermum australe* exhibit a strong glycosidase inhibition [229,230,231] and show promising potential to be used as chemotherapeutic [219], antiviral [228], and immunomodulatory drugs.

A pair of PAs with antitumor as well as antibiotic properties are (-)-clazamycin A and (+)-clazamycin B, which were isolated from *Streptomyces puniceus* [232,233]. These compounds show activity against *Herpes simplex* virus in vitro [233], as well as a strong inhibition of *Pseudomonas* species [232]. Moreover, in some fungus species, PAs with potent activity against Gram-positive and Gram-negative bacteria, as well as tumor cells have been isolated [234,235].

## 6. Occurrence of Pyrrolizidine Alkaloids in Crop Plants

While many plant species are PA producers, few of them are used by humans, which is likely related to PA toxicity. Among them, most are used in traditional medicine for treatment of diverse diseases, which is discussed in detail in a number of review articles [16,17,177,236,237,238,239]. However, there are also several PA-producing plants used as food or as forage crops. While intake of medicinal plants is usually minute, vegetables and herbs may be consumed at substantial amounts, making presence of PAs very critical. Thus, we will focus here on plant species that are consumed as food or used as livestock feed in different regions of the world. 

### 6.1. Borago officinalis

Borage (*Borago officinalis*), which originated from the Mediterranean area, is a common annual garden herb used in salads and teas and is widely cultivated for seed oil production. The oil contains high amounts of the essential fatty acid γ-linolenic acid (18–25%), and is therefore used for nutritional, medical, and cosmetic purposes. Larson et al. [240] first identified the pyrrolizidine alkaloids lycopsamine and amabiline (Figure 17) in fresh and dried leaves and stems of *B. officinalis* with a total alkaloid amount of less than 0.001% dry weight. They also showed that in roots the alkaloids are mainly present as the free base, while fresh leaves contain mainly *N*-oxides. Lüthy et al. [241] did not only detect lycopsamine and amabiline but also intermedine, supinine and the isomer pair acetyllycopsamine/acetylintermedine in the range of 2-10 mg/kg in vegetative tissues. Dodson et al. [92] determined thesinine as the only alkaloid in the flowers and the major alkaloid in seeds. Mature seeds were found to contain thesinine and minute amounts of lycopsamine. Immature seeds contained only thesinine. The first glycosylated pyrrolizidine alkaloid reported in plants, namely thesinine-4′-*O*-β-d-glucoside, was discovered in dried and defatted borage seeds [53]. Seeds contain approximately 0.1% to 0.5% thesinine-4′-*O*-β-d-glucoside. 

Wretensjö and Karlberg [242] analyzed crude borage oil as well as borage oil from different processing stages and found no pyrrolizidine alkaloids present above a detection limit of 20 µg/kg. Additionally, they could demonstrate by spiking experiments with crotaline that the pyrrolizidine content in crude borage oil was reduced overall by a factor of approximately 30,000 in the refining process. Additionally, Vacillotto et al. [243] could not detect any pyrrolizidine alkaloids above a detection limit of 0.2 µg/kg in seed oil.

### 6.2. Crassocephalum crepidioides

The genus *Crassocephalum* belongs to the Asteraceae family (subfamily Asteroideae, tribe Senecioneae) [244] and includes approximately 100 plant species growing in Africa, tropical and subtropical regions of Asia and Australia and locally in America [245,246]. Both *Crassocephalum crepidioides* (Figure 18A) and *Crassocephalum rubens* are erect annual herbs used as leafy vegetables [245,247,248,249].

*C. crepidioides* and *C. rubens* are used fresh or dried for preparation of sauces, soups and stews [250]. *C. crepidioides* is also eaten raw, for example in Australia in salad dishes [245]. A factor making particularly *C. crepidioides* interesting for food production is its high yield of up to 25-27 t/ha per year even on soils with low nutrient supply [245]. Moreover, it has desirable nutritional properties since dried leaves contain about 27% protein, 8% fiber, 42% carbohydrates, and 3% lipids [248,251]. These values suggest that *C. crepidioides* is a good source of fiber while the relatively high amount of carbohydrates makes it a good source of energy. *C. crepidioides* and *C. rubens* contain also high levels of minerals including potassium, magnesium, calcium, phosphorous, iron, copper and manganese [251]. Both plant species are also used in traditional medicine due to supposed medicinal properties, such as antibiotic, antioxidant, anti-fungal, anti-inflammatory, anti-diabetic and blood regulative effects. They are included in traditional treatment of hepatic insufficiency, intestinal worms, stomach disorders and open wounds [245,252,253]. Furthermore, in Africa *C. crepidioides* is traditionally part of diets of pregnant and breast-feeding women as it is believed to prevent anemia and stimulate milk production [252]. In addition to human consumption and application in traditional medicine, *C. crepidioides* is also used as livestock feed, for instance for poultry [245]. 

Based on brine shrimp lethality tests and application of Mayer’s reagent Adjatin et al. considered both *C. crepidioides* and *C. rubens* as safe for human consumption [254]. In sharp contrast, Asada et al. reported isolation of significant amounts of jacobine and jacoline (Figure 18C) from *C. crepidioides* [255]. The reason for these divergent results might be the low sensitivity of Mayer’s reagent for PAs, which are in fact usually detected with Dragendorff’s reagent, or with higher sensitivity and specificity, by chromatographic techniques. Indeed, presence of high amounts of jacobine in *C. crepidioides* was recently confirmed by cation exchange chromatography [50]. Interestingly, the content of jacobine depended highly on the tissue. In young leaves jacobine was present at levels exceeding 200 mg/kg fresh weight while it was below 1 mg/kg fresh weight in old leaves. In addition, the jacobine content was under developmental control since six-week-old plants showed contents of approximately 27 mg/kg fresh weight while 10-week-old plants had contents of 260 mg/kg. Also, the growth conditions had a significant impact since incubator-grown plants had high levels while jacobine was below the detection limit (1 mg/kg) in greenhouse grown plants [50]. Production of alkaloids has also been reported for *C. rubens* [256]. These results clearly show that *C. crepidioides* and *C. rubens* cannot be considered safe for human consumption and that PA-free varieties have to be identified prior to utilization of these species as food or forage crops.

### 6.3. Gynura

*Gynura*, a genus of the Asteraceae family, contains several edible species native to Asia like *G. bicolor*, *G. divaricata* and *G. procumbens* which are used in Asian cuisine and traditional medicine. The species *G. bicolor* for example, has great nutraceutical potential, containing high levels of phenolic acids, flavonoids, carotenoids and anthocyanins and exhibits anti-oxidative as well as anti-inflammatory properties [257]. Furthermore, it can be used for the treatment of diabetes mellitus [258] and cancer [259,260]. Even though the use of *Gynura* may have numerous beneficial effects on health, its consumption is not without risk because of the presence of PAs. Intergerrimine and usaramine were first identified in *G. divaricata* by Roeder et al. [261]. Chen et al. [262] compared the PA profiles of *G. bicolor* and *G. divaricata* collected from five different provinces of China. They found 27 PAs consisting of five different types in the two species. Known PAs that were identified are retrorsine, spartioidine, seneciphylline, integerrimine, senecionine, senkirkine, retrorsine *N*-oxide, senecionine *N*-oxide, and seneciphylline *N*-oxide (Figure 19). Additionally, 18 other unidentified PAs were detected in both species. *G. divaricata* collected in the Jiangsu province showed the highest concentration (39.7 µg/g) of retronecine type PAs while *G. bicolor* from Fujian contained the lowest concentration (1.4 µg/g). *G. divaricata* always contained higher amounts of retronecine type PAs in comparison to *G. bicolor* from the same locality and harvest time. Both species collected from Jiangsu contained the most diverse PA profile, while *G. divaricata* from Fujian showed the lowest variety of PAs. A weak cytotoxic activity was detected in alkaloid extracts of both species collected from Jiangsu. These results show that the amount and variety of PAs in *Gynura* depends on the species as well as the growing region, indicating that environmental factors likely play a role in PA accumulation in *Gynura*. Another *Gynura* species traditionally used in Chinese medicine is *Gynura segetum*, also known as Jusanqi. It has been shown to have anti-angiogenic [263] as well as antioxidative and anti-inflammatory properties [264]. Its medicinal use is risky though as there have been several reports of hepatic veno-occlusive disease linked to the consumption of *G. segetum* [265,266,267,268]. Liang and Roeder were the first to characterize senecionine from *G. segetum* [269]. The presence of senecionine was confirmed by Yuan et al. [270], who could additionally detect seneciphylline as well as seneciphyllinine and *E*-seneciphylline. Qi et al. [271] analyzed whole plants of *G. segetum* collected from Zhejang province of China and could unequivocally identify seneciphylline, senecionine, seneciphyllinine, and seneciphyllinine *N*-oxide using HPLC / ITMS. 16 further PA or PANO structures were tentatively assigned, among them *E*-seneciphylline, *E*-seneciphylline *N*-oxide, senecicannabine, senecicannabine *N*-oxide, a yamataimine isomer, jacoline, and tetrahydrosenecionine, which has been reported for the first time from a natural source. Considering the present research about PAs in *Gynura* indicates that caution is required concerning its use for dietary and medicinal purposes.

### 6.4. Lolium

*Lolium* species, commonly known as ryegrass, are important pasture grass species serving as main feed for grazing livestock in numerous countries and regions, for example Australia, New Zealand, North America and Europe. These species can have toxic effects on mammals that graze them and are related to a syndrome known as ryegrass staggers. Therefore, their chemical composition has been analyzed intensively.

Interestingly, the originally identified toxins, lolitrem B [272], ergovaline [273], and loline alkaloids are produced by endophytic fungi [5]. The loline alkaloids represent a special class of PAs, which are characterized by the presence of an ether bridge (Figure 1). In contrast to classical PAs, which are produced by plants, lolines are metabolic products of fungi, particularly of the genus *Epichloë* [5,274,275]. 

In addition, several *Lolium* species can also produce PAs of the thesinine type themself. Koulman et al. [54] were the first to isolate and elucidate the stereoisomers *E*-thesinine-*O*-4′-α-rhamnoside and *Z*-thesinine-*O*-4′-α-rhamnoside from *Lolium perenne* (Figure 20), thereby providing the first report of PAs in the Poaceae family which are synthesized by the plant itself. Furthermore, they could detect those two compounds together with the aglycone and a hexoside in *Festuca arundinacea*, a species of grasses closely related to *L. perenne*. In *L. perenne* the highest concentrations of *E*/*Z*-thesinine-*O*-4′-α-rhamnoside were found in sheath tissue, followed by blade tissue while immature tissue contained the lowest amounts. Infection with the endophytic fungus *Neotyphodium lolii* increased the concentrations only slightly. The authors analyzed plants grown in a greenhouse and found variations in the ratios of *E*-thesinine-*O*-4′-α-rhamnoside to *Z*-thesinine-*O*-4′-α-rhamnoside between 0.2 and 1. When clonal material was grown in a climate room under controlled conditions the variation was severely reduced, indicating a strong environmental influence on the accumulation of *E*/*Z*-thesinine-rhamnoside. Since the identified alkaloid conjugates do not possess a C-1/C-2 unsaturation and a hydroxy group at C-7, they likely exhibit a limited toxicity to mammals. The common occurrence of these PAs in several widely used commercial *L. perenne* and *F. arundinacea* cultivars further demonstrates their non-toxicity or low toxicity to grazing livestock, but detoxification by rumen bacteria might play a role as well [276]. It is currently unknown if the thesinine-rhamnoside PAs play any role in insect deterrence or if they enhance plant fitness by any other means. Wesseling et al. [91] analyzed various Pooideae species grown under controlled conditions and found that PA biosynthesis is restricted to a small group of species producing only a narrow range of thesinine conjugates. The three closely related outbreeding *Lolium* species *L. perenne*, *L. multiflorum*, and *L. rigidum* were shown to accumulate PAs while the inbreeding *L. temulentum* and *L. remotum* do not. Furthermore, *F. arundinacea* contained PAs while in the closely related *F. pratensis* they were absent. Potential explanations for this pattern might be the independent evolution of PA biosynthesis in *F. arundinacea* or a secondary loss in the inbreeding *Lolium* species. Hybridization of PA producing *Lolium* species with *Festuca* could be another possibility of how PA production was acquired in *Festuca*. Additionally, the inter and intra-specific PA patterns are highly variable. *L. rigidum* contained the highest average amounts of PAs, followed by *L. perenne* and *F. arundinacea*, while *L. multiflorum* showed the lowest concentrations. *F. arundinacea* was the only species which contained the aglycone thesinine in significant amounts. In *L. perenne* and *L. multiflorum* PAs were restricted to only a few of the tested individuals, while all samples of *L. rigidum* and *F. arundinacea* were tested positive. It seems like PA biosynthesis is a constitutive trait in some species, while in others it occurs only infrequently and might possibly be induced by external stimuli or occur in later developmental stages. Gill et al. [76] identified two putative HSS genes in perennial ryegrass, one of them (*LpHSS1*) occurring only sporadically. A significant association of absence of the *LpHSS1* gene with lower levels of thesinine-rhamnoside PAs was found. This provides the possibility of developing genetic markers for future breeding efforts in regard to PA presence and enables investigations about the role of PAs in deterring herbivore pests. Further HSS-like gene sequences were identified in other Poaceae species, including tall fescue, wheat, maize, and sorghum. Therefore, PA biosynthesis might be even more widespread in the grass family than it is known to date with unknown PA structures still to be discovered.

## Figures and Tables

**Figure 1 molecules-24-00498-f001:**
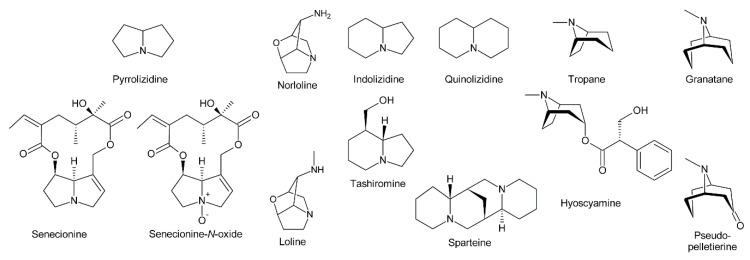
Core structures and examples for pyrrolizidine, loline, indolizidine, quinolizidine, tropane and granatane alkaloids. In contrast to the other alkaloids pyrrolizidine alkaloids appear mainly as *N*-oxides, as shown for the example of senecionine-*N*-oxide.

**Figure 2 molecules-24-00498-f002:**
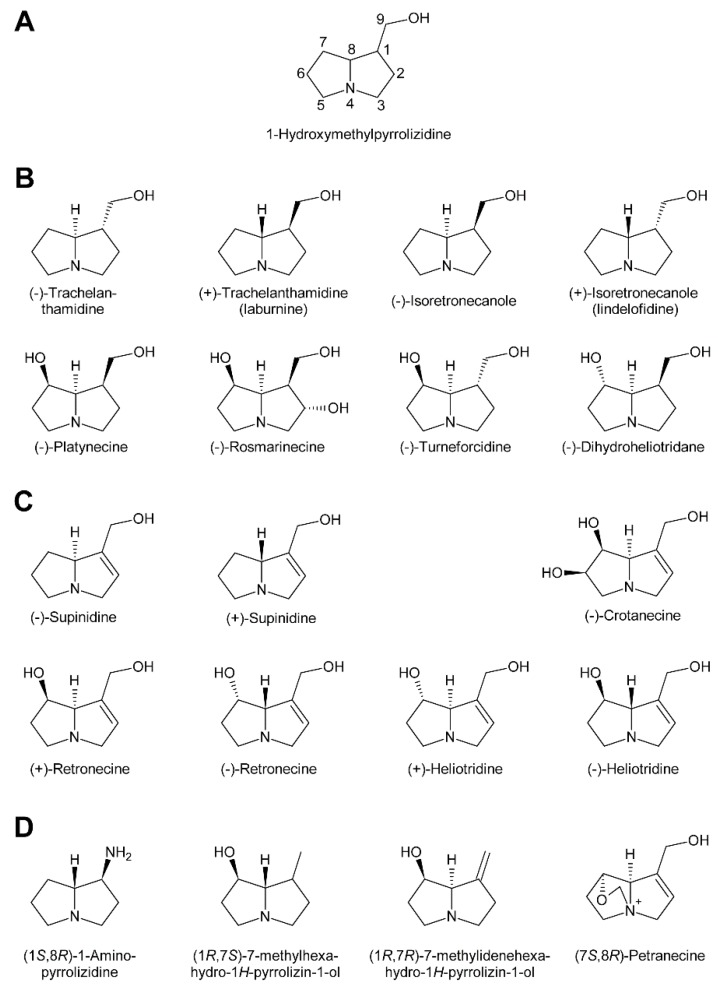
Structures of necine bases. (**A**) Basic structures and numbering of atoms in necine bases. (**B**) Saturated necine bases. (**C**) 1,2-Desaturated necine bases. (**D**) Unusual necine bases.

**Figure 3 molecules-24-00498-f003:**
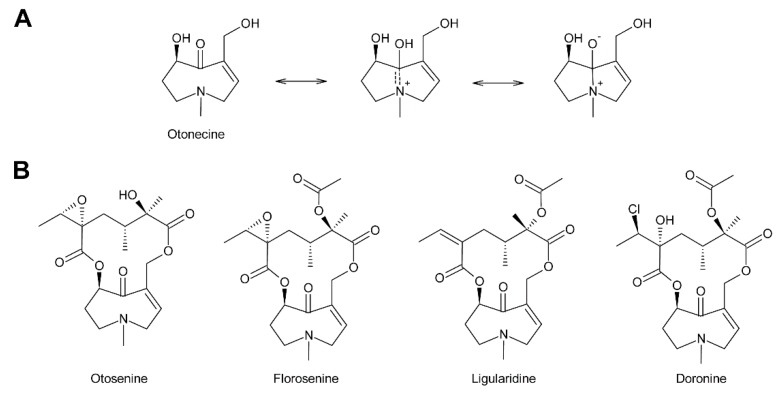
Otonecine and representative PAs. (**A**) Resonance structures of otonecine. (**B**) Structures of the otonecine-type PAs otosenine, florosenine, ligularidine, and doronine.

**Figure 4 molecules-24-00498-f004:**
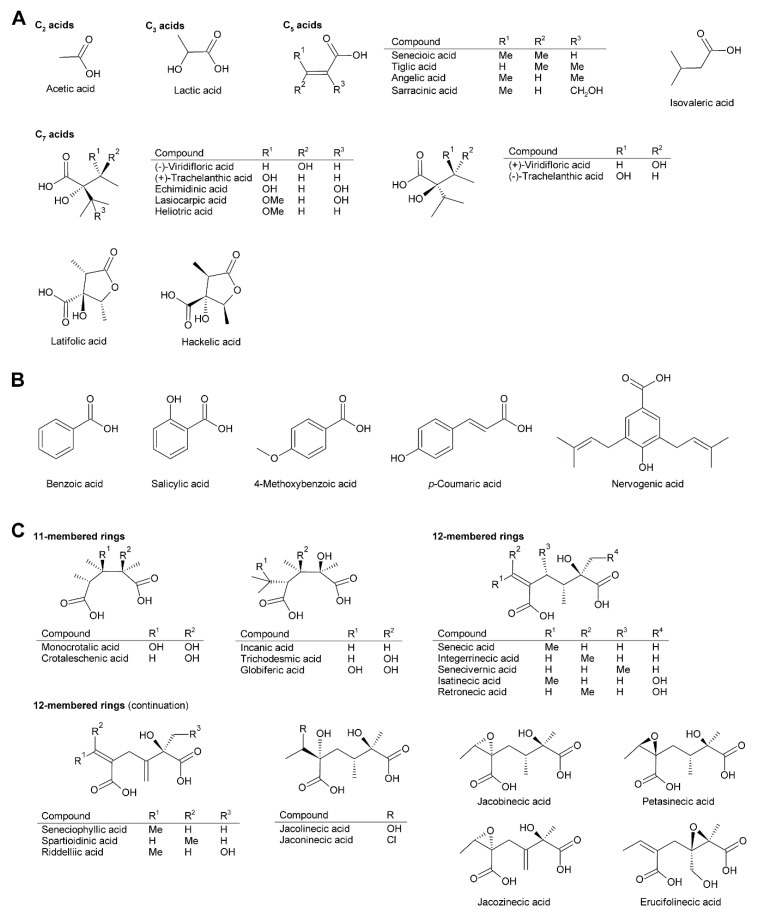
Examples for necic acids. (**A**) Monocarboxylic aliphatic acids. (**B**) Monocarboxylic aromatic acids. (**C**) Dicarboxylic acids forming macrocyclic PAs. Adapted from Reference [17].

**Figure 5 molecules-24-00498-f005:**
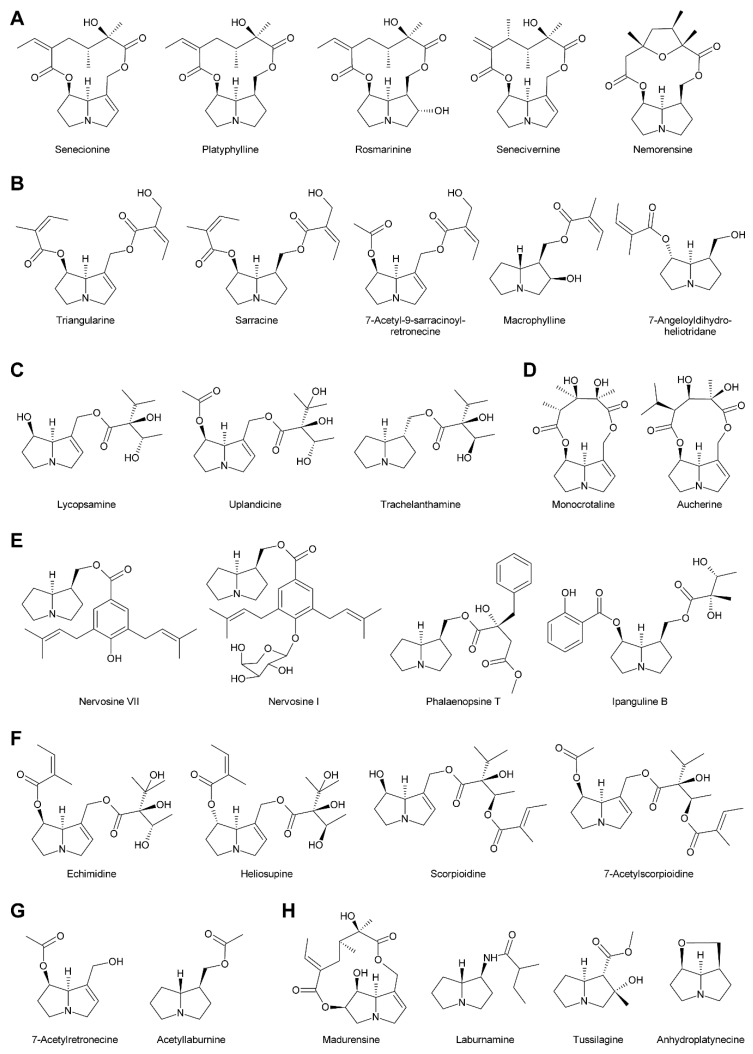
Linkage patterns of necic acids with necine bases. (**A**) Senecionine type. (**B**) Triangularine type. (**C**) Lycopsamine type. (**D**) Monocrotaline type. (**E**) Phalaenopsine/ipanguline type. (**F**) Compounds combining necic acids of triangularine and lycopsamine types. (**G**) Simple PAs. (**H**) PAs with unusual linkage patterns.

**Figure 6 molecules-24-00498-f006:**
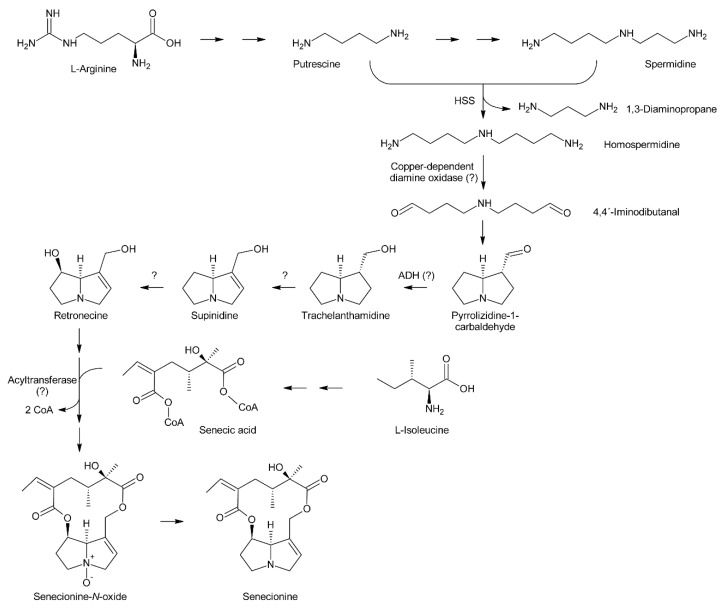
Biosynthesis of PAs. The polyamines putrescine and spermidine are derived from the basic amino acid arginine. Subsequently, homospermidine synthase (HSS) exchanges the 1,3-diamonopropane residue of spermidine by putrescine, which releases 1,3-diaminopropane and forms symmetric homospermidine. Oxidation of homospermidine, likely by copper-dependent diamine oxidases, to 4,4´-iminodibutanal initiates cyclization to pyrrolizidine-1-carbaldehyde, which is reduced, likely by an alcohol dehydrogenase, to 1-hydroxymethylpyrrolizidine. Desaturation and hydroxylation by unknown enzymes form retronecine, which is acylated with an activated necic acid, for instance with senecyl-CoA_2_ as in the example shown above. Acylation might be catalyzed by an acyltransferase of the BAHD family. PA *N*-oxides, which are believed to be the primary products of PA biosynthesis, may be reduced to the free tertiary amine.

**Figure 7 molecules-24-00498-f007:**
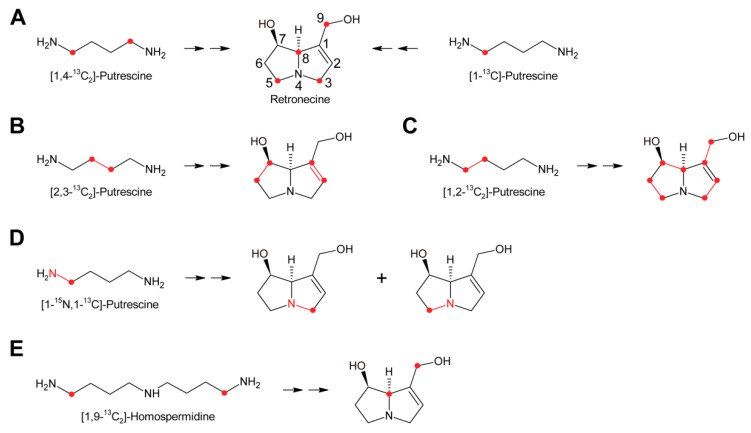
Investigation of necine base biosynthesis by feeding of stable isotope labeled precursors. Feeding of (**A**) [1,4-^13^C_2_]-putrescine and [1-^13^C]-putrescine, (**B**) [2,3-^13^C_2_]-putrescine, (**C**) [1,2-^13^C_2_]-putrescine, (**D**) [1-^15^N, 1-^13^C]-putrescine and (**E**) [1,9-^13^C_2_]-homospermidine. Red dots symbolize ^13^C labels, a red N symbolizes a ^15^N label. ^13^C-^13^C double labels are marked with bonds in red. Please mind that the retronecine structures shown in (A), (B), and (C) are composite representations of all labeled species present.

**Figure 8 molecules-24-00498-f008:**
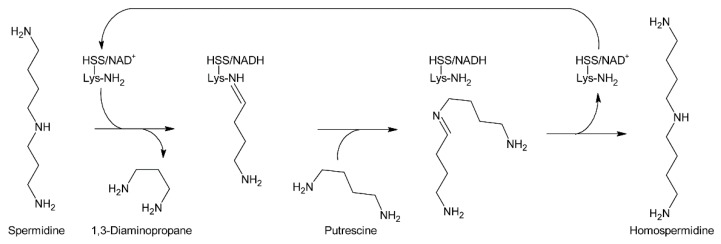
Mechanism of homospermidine formation by HSS. The amino group of a lysine residue of HSS reacts with spermidine, which releases 1,3-diaminopropane and reduces HSS-bound NAD^+^ to NADH. Next, the residue is transferred to putrescine, forming an imine intermediate, which is reduced by the HSS-bound NADH to release homospermidine and recycle the HSS/NAD^+^ complex. Adapted from [72].

**Figure 9 molecules-24-00498-f009:**
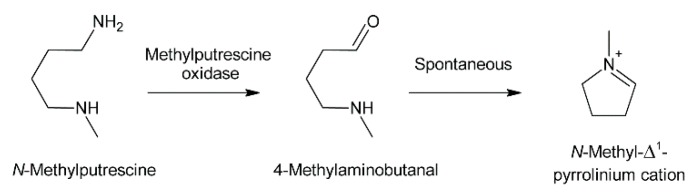
Cyclization in tropane alkaloid biosynthesis. The precursor *N*-methylputrescine is oxidized by methylputrescine oxidase to 4-methylaminobutanal, which cyclizes spontaneously by reaction of the amino group with the aldehyde. Adapted from Reference [9].

**Figure 10 molecules-24-00498-f010:**
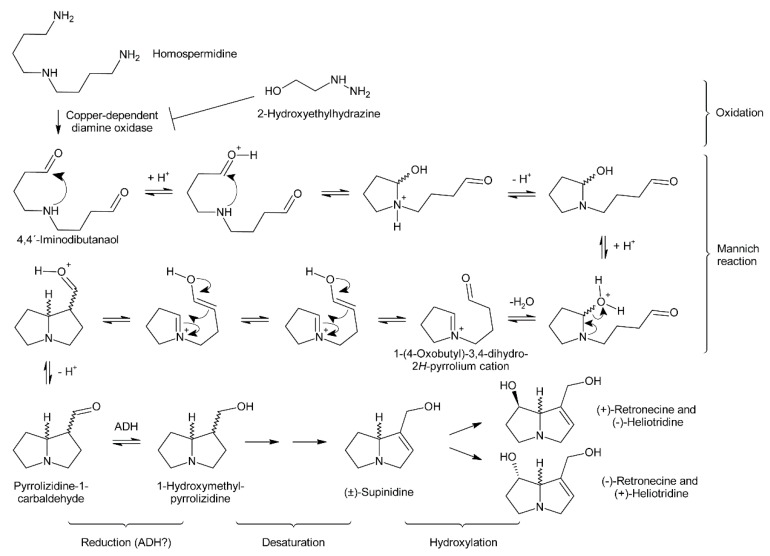
Cyclization in PA biosynthesis. Homospermidine is likely oxidized by a copper-dependent diamine oxidase to 4,4′-iminodibutanal, which can be inhibited by the synthetic compound 2-hydroxyethylhydrazine. The reaction product cyclizes first to the 1-(4-oxobutyl)-3,4-dihydro-2*H*-pyrrolium cation and further to pyrrolizidine-1-carbaldedye in a Mannich-type reaction. However, as indicated, spontaneous cyclization would lead to a mixture of the different stereomers and thus mixtures of necine bases would be obtained, arguing for enzyme-catalyzed cyclization.

**Figure 11 molecules-24-00498-f011:**
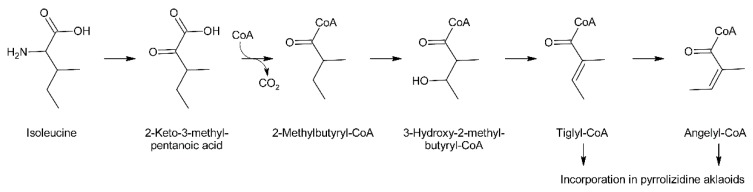
Biosynthesis of tiglic and angelic acid. l-Isoleucine is desaminated and the obtained ketocarboxylic acid decarboxylated to 2-methylbutyric acid, which is likely accompanied by linking with coenzyme A to yield 2-methylbutyryl-CoA. This intermediate is hydroxylated and subsequently dehydrated to form tiglyl-CoA, which can be isomerized to angelyl-CoA. The activated tiglic and angelic acid residues are finally transferred to necine bases and coenzyme A is released.

**Figure 12 molecules-24-00498-f012:**
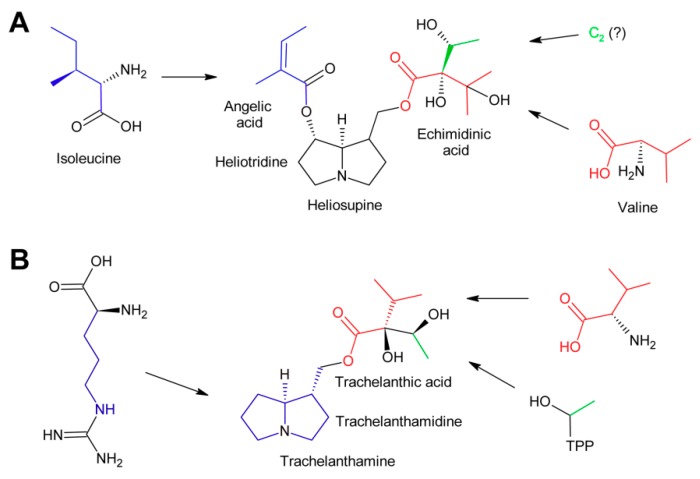
Biosynthesis of C_7_ necic acids. (**A**) Echimidinic acid present in heliosupine is derived from l-valine (red) and a C_2_ unit of unknown origin (green). Angelic acid esterifying the 7-hydroxy group of (-)-heliotridine is derived from l-isoleucine (blue) as shown in Figure 11. (**B**) Trachelanthic acid is also derived from l-valine (red) and a C_2_ unit (green) that is likely derived from hydroxyethyl thiamine pyrophosphate. Adapted from References [70,103].

**Figure 13 molecules-24-00498-f013:**
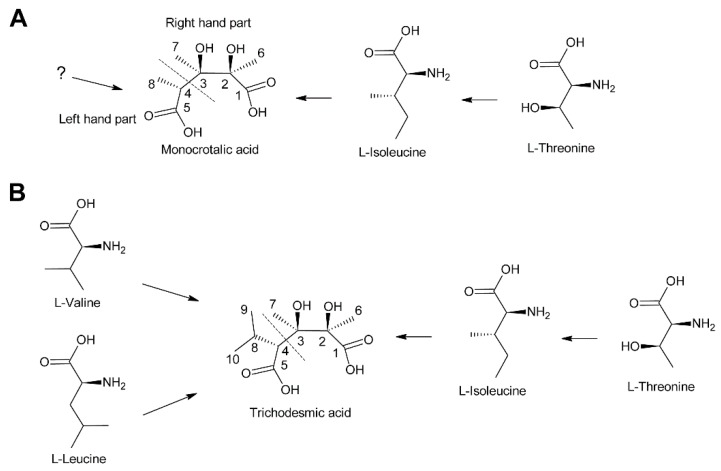
Biosynthesis of necic acids forming 11-membered rings. (**A**) The right hand part of monocrotalic acid is formed from l-leucine (and its precursor l-threonine). The C_3_ left hand unit is of unknown origin. (**B**) The right hand side of trachelanthic acid is derived from l-leucine (and its precursor l-threonine) while the left hand side seems to be formed from l-valine or l-leucine. Dotted lines indicate the both sides. Adapted from Reference [70].

**Figure 14 molecules-24-00498-f014:**
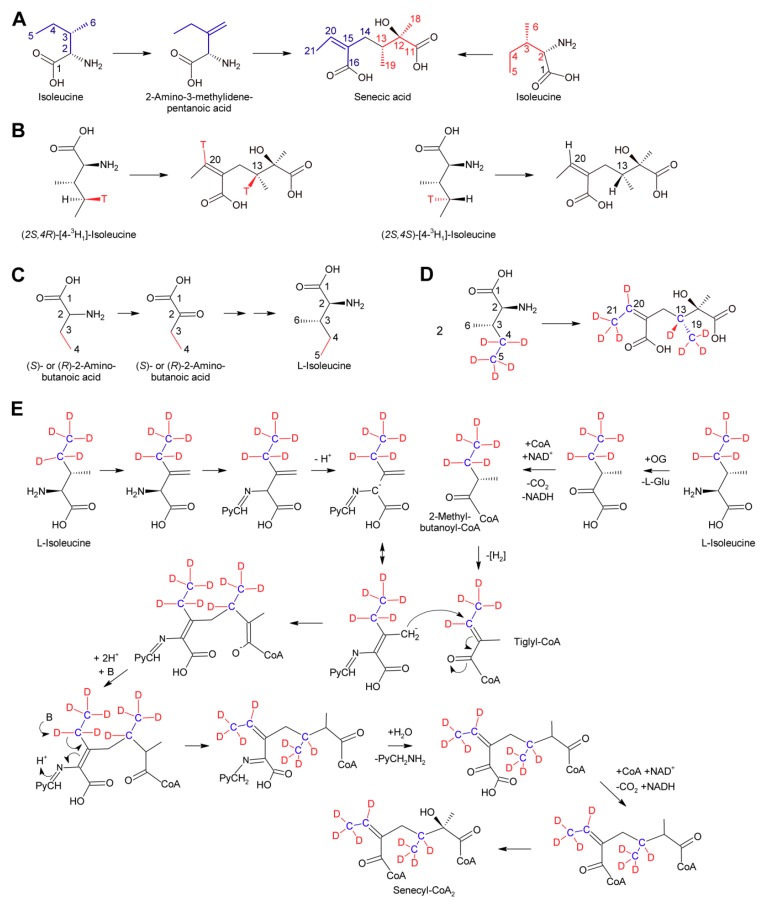
Biosynthesis of senecic acid. (**A**) Feeding experiments with ^14^C-labeled compounds showed that senecic acid is formed from two l-isoleucine molecules accompanied by loss of both carboxy carbons. 2-Amino-3-methylenepentanoic acid might be an intermediate, although it is not clear whether for one or both halves of senecic acid. Carbons are numbered according to Reference [112]. (**B**) Feeding with l-isoleucine stereospecifically labeled with ^3^H (tritium, T) at C-4 showed that only the *4R* label was retained upon incorporation. (**C**) 2-Aminobutanoic acid is converted *in planta* to l-isoleucine. Due to 2-ketobutanoic acid as intermediate the initial stereochemistry of 2-aminobutyric acid is irrelevant. A label (^13^C or ^3^H; red) at C-3 and C-4 is retained and found on position C-5 and C-6 in l-isoleucine. (**D**) The ^13^C-labeled C-4 and C-5 (blue) of l-leucine (obtained in planta from labeled 2-aminobutanoic acid) and most of the ^2^H (deuterium, D; red) is retained upon incorporation into senecic acid. (**E**) Possible mechanism for formation of senecic acid. CoA, coenzyme A; l-Glu, l-glutamic acid; NAD^+^/NAHD, oxidized/reduced nicotinamide adenine dinucleotide; OG, 2-oxoglutaric acid; and Py, pyridoxalphosphate.

**Figure 15 molecules-24-00498-f015:**
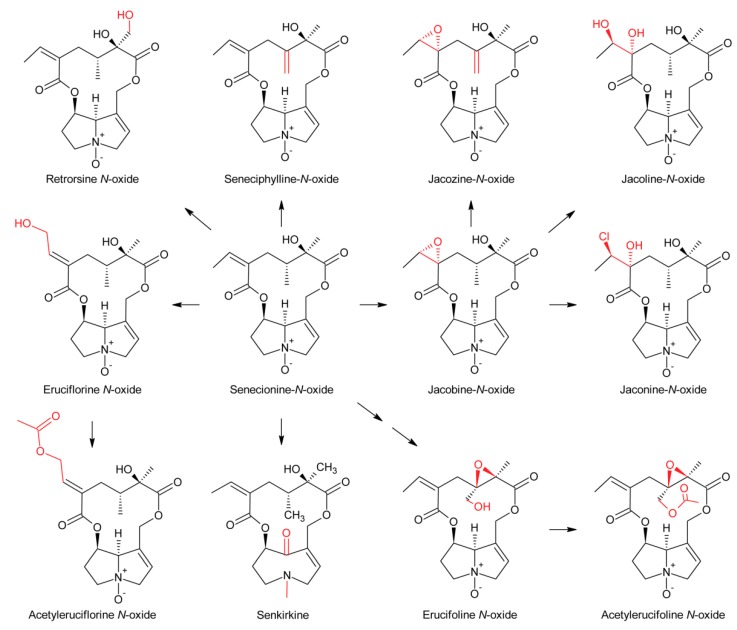
Diversification of senecionine-like PAs. Modifications of the senecionine *N*-oxide structure are shown in red. Adapted from Reference [116].

**Figure 16 molecules-24-00498-f016:**
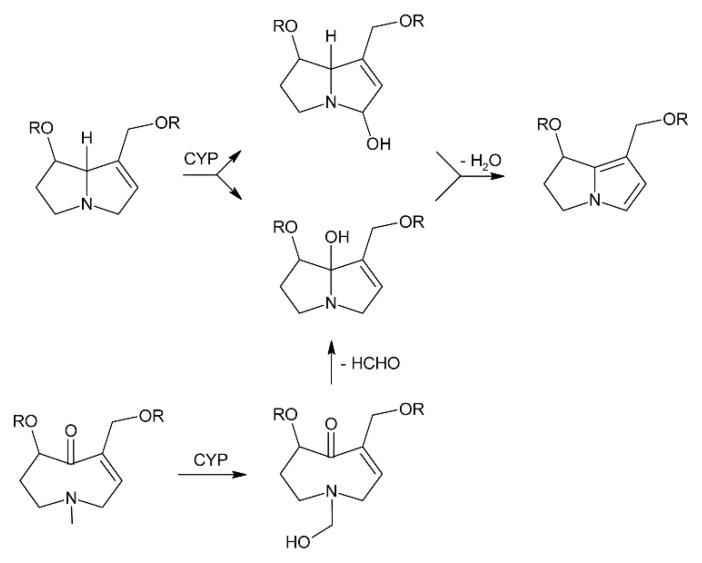
Metabolic activation of pyrrolizidine alkaloids. PAs are oxidized by cytochrome P450 enzymes (CYP), which converts them to the reactive and toxic pyrrole derivatives. Adapted from Reference [11].

**Figure 17 molecules-24-00498-f017:**
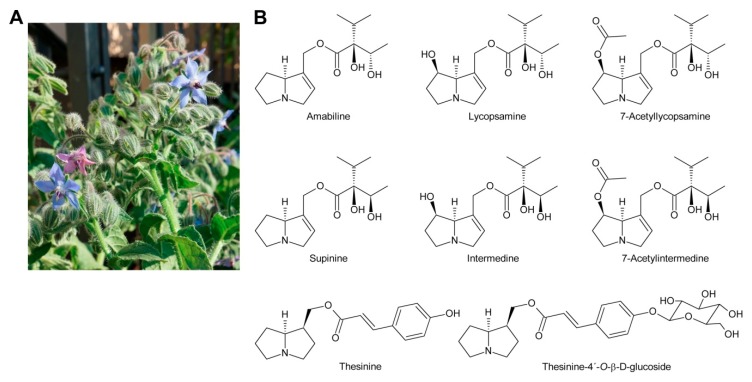
Pyrrolizidine alkaloids found in *Borago officinalis* (borage): (**A**) Flower of *B. officinalis*. Young flowers show a bright blue color while older flowers exhibit, because of a lower pH, a slightly reddish coloration. (**B**) Leaves of *B. officinalis* contain the supinidine-type PAs amabiline and supinine and the retronecine-type lycopsamine and intermedine as well as their 7-acetylated derivatives. Seeds contain mainly the (-)-isoretronecanol-type PA thesinine and its glucoside thesinine-4′-*O*-*β*-d-glucoside.

**Figure 18 molecules-24-00498-f018:**
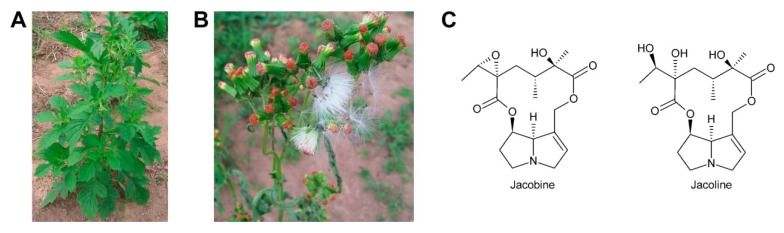
*Crassocephalum crepidioides*. *C. crepidioides* plant (**A**) and flowers (**B**). (**C**) The retronecine-type PAs jacobine and jacoline were reported in *C. crepidioides*. Pictures shown in (**A**) and (**B**): Courtesy of N. Adebimpe Adedej.

**Figure 19 molecules-24-00498-f019:**
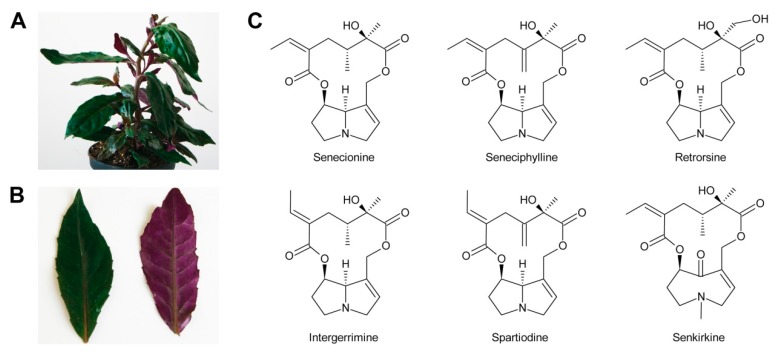
Pyrrolizidine alkaloids observed in *Gynura bicolor*. (**A**) *G. bicolor* plant. (**B**) *G. bicolor* leaves from the upper (green) and lower (ruby-colored) side. (**B**) Senecionine, seneciphylline and retrorsine were reported in *G. bicolor* in both, their free forms (shown above) and as *N*-oxides, while intergerrimine, spartiodine, and senkirkine were only found as free alkaloids.

**Figure 20 molecules-24-00498-f020:**
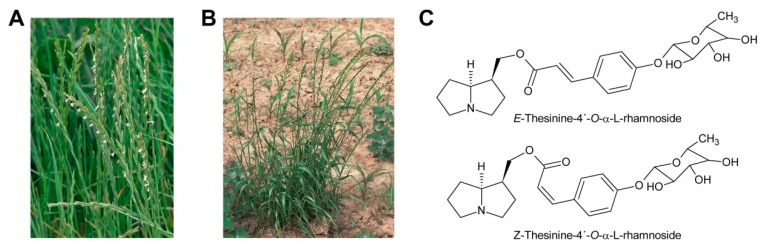
Pyrrolizidine alkaloids reported for *Lolium* species. (**A**) *Lolium perenne* (ray grass). (**B**) *Lolium multiflorum* (Italian rye-grass). (**C**) Structures of *E-* and *Z-*thesinine-4′-*O*-α-l-rhamnoside. Pictures shown in (A) and (B): Courtesy of Leo Michels.

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
