# Peer review of "Pyrrolizidine Alkaloids: Biosynthesis, Biological Activities and Occurrence in Crop Plants"

_molecules, 2019, doi:10.3390/molecules24030498_

Round 1
Reviewer 1 Report
This is a well-written and very extensive literature review about pyrrolizidine alkaloids. Almost every aspect is covered, from biosynthesis to functions, and the authors carefully explain what is known and still unknown about these plant secondary metabolites.
Although there is a large section on biosynthesis, there is nothing on the heritability of alkaloid concentrations or compositions in plants which is worth spending a few lines on.
The last section on Lolium and the loline alkaloids is unclear. First the authors state that these alkaloids are produced by endophytes, later they discuss PA production of different grasses and genes (plant or fungal?) involved in production. So is it the fungi or the plants that produce these alkaloids? If both, please clarify.
There are many figures in this review. Some (e.g. fig. 6, fig. 15) look rather similar to figures from earlier papers, and it would be appropriate to cite these references in the figure legends. From example as "modified after..." All relevant papers are cited in the main text.
Figure 5 is cited in the text before Figure 3 and 4.
l.239-240 reference is missing.
l.1016-1028 too long paragraph only describing plant species without reference to alkaloid content.
Author Response
Response to Reviewer 1 Comments
Reviewer 1
This is a well-written and very extensive literature review about pyrrolizidine alkaloids. Almost every aspect is covered, from biosynthesis to functions, and the authors carefully explain what is known and still unknown about these plant secondary metabolites.
Response:
Many thanks for the very positive evaluation of our manuscript!
Comment 1:
Although there is a large section on biosynthesis, there is nothing on the heritability of alkaloid concentrations or compositions in plants which is worth spending a few lines on.
Response 1:
We agree that heritability of PA levels is an important topic that was not covered in the previous version. We have now included a paragraph about heritability in chapter 4 (lines 691 to 723 of the revised manuscript).
Comment 2:
The last section on Lolium and the loline alkaloids is unclear. First the authors state that these alkaloids are produced by endophytes, later they discuss PA production of different grasses and genes (plant or fungal?) involved in production. So is it the fungi or the plants that produce these alkaloids? If both, please clarify.
Response 1:
Occurrence of PAs in Lolium is very interesting since on one hand lolines produced by entophytic fungi and on the other hand PAs, particularly thesinine glycosides, produced by the plant have been described.
We agree that the presentation was not clear in the previous version. We have clarified that now (lines 1132 to 1143 of the revised manuscript).
Comment 3:
There are many figures in this review. Some (e.g. fig. 6, fig. 15) look rather similar to figures from earlier papers, and it would be appropriate to cite these references in the figure legends. From example as "modified after..." All relevant papers are cited in the main text.
Response 3:
Several figures were indeed modified from previously published manuscripts. We agree that the corresponding source should be mentioned directly in the reference and not only in the main text. Thus, we have now included the references.
However, Figure 6 was not adapted from a certain previously published manuscript. Thus, we did not include a specific reference for that figure.
Comment 4:
Figure 5 is cited in the text before Figure 3 and 4.
Response 4:
We forgot to mention Figures 2, 3 and 4 in lines 64 to 65. Now they are included and thus the correct order is established.
Comment 5:
l.239-240 reference is missing.
Response 5:
Indeed, we forgot to include a number of references in the paragraph from line 233 to line 241. We have included those references now.
Comment 6:
l.1016-1028 too long paragraph only describing plant species without reference to alkaloid content.
Response 6:
In contrast to the other plant species mentioned in chapter 6, Crassocephalum crepidioides is largely unknown. Thus, we wanted to provide some background information about that species. However, we agree that we included too many details. We have now shortened the description and mention only the most important aspects (lines 1050 to 1054 of the revised manuscript).
Reviewer 2 Report
Evaluation and comments to the manuscript No. molecules-421423 entitled “Pyrrolizidine Alkaloids: Biosynthesis, Biological Activities and Occurrence in Crop Plants”.
Authors: Sebastian Schramm, Nikolai Köhler, Wilfried Rozhon
The authors of the manuscript present in a very mature and high level the issues related to pyrrolizidine alkaloids (PAs) which are heterocyclic secondary metabolites with a typical pyrrolizidine motif predominantly produced by plants as defense chemicals against herbivores. Therefore, in my opinion, the work may be of interest to the readership of the Molecules journal. Moreover, it may be well cited in the future.
Overall, the MS is written in correct scientific language. I did not notice any major factual or editorial errors at work. On the other hand, I do not mention any minor errors, because they are in some sense something normal in this type of work and they do not affect the general perception of it. I would like to emphasize once again the high level of scientific and technical work. Therefore, I have no serious substantive comments to the MS, and in my opinion the work should make a good contribution to the literature.
My conclusion
In my opinion, the work should be accept and published.
Author Response
Response to Reviewer 2 Comments
Reviewer 2
Evaluation and comments to the manuscript No. molecules-421423 entitled “Pyrrolizidine Alkaloids: Biosynthesis, Biological Activities and Occurrence in Crop Plants”.
Authors: Sebastian Schramm, Nikolai Köhler, Wilfried Rozhon
The authors of the manuscript present in a very mature and high level the issues related to pyrrolizidine alkaloids (PAs) which are heterocyclic secondary metabolites with a typical pyrrolizidine motif predominantly produced by plants as defense chemicals against herbivores. Therefore, in my opinion, the work may be of interest to the readership of the Molecules journal. Moreover, it may be well cited in the future.
Overall, the MS is written in correct scientific language. I did not notice any major factual or editorial errors at work. On the other hand, I do not mention any minor errors, because they are in some sense something normal in this type of work and they do not affect the general perception of it. I would like to emphasize once again the high level of scientific and technical work. Therefore, I have no serious substantive comments to the MS, and in my opinion the work should make a good contribution to the literature.
My conclusion
In my opinion, the work should be accept and published.
Response:
Many thanks for the very positive evaluation of our manuscript!
Reviewer 3 Report
The authors present a detailed work on the topic of PA, mainly chemistry, biosyhnthesis and toxicological aspects.
The problem is that a very similar paper has been published last year by International Journal of Molecular Sciences:
Pyrrolizidine Alkaloids: Chemistry, Pharmacology, Toxicology and Food Safety
Int. J. Mol. Sci. 2018, 19(6), 1668; https://doi.org/10.3390/ijms19061668
The topic of this submission is largely overlapping with this already published paper that, altough cited, had already discussed the major topics regarding chemistry, biosynthesis and toxicity.
For this very important reason, i believe the current submission lacks enough novelty to be published.
Author Response
Response to Reviewer 3 Comments
Reviewer 3
The authors present a detailed work on the topic of PA, mainly chemistry, biosyhnthesis and toxicological aspects.
Response:
We appreciate that reviewer 3 considers our manuscript as a detailed work about PAs.
Comment 1:
The problem is that a very similar paper has been published last year by International Journal of Molecular Sciences:
Pyrrolizidine Alkaloids: Chemistry, Pharmacology, Toxicology and Food Safety
Int. J. Mol. Sci. 2018, 19(6), 1668; https://doi.org/10.3390/ijms19061668
The topic of this submission is largely overlapping with this already published paper that, altough cited, had already discussed the major topics regarding chemistry, biosynthesis and toxicity.
For this very important reason, i believe the current submission lacks enough novelty to be published.
Response 1:
We are surprised that reviewer 3 thinks that our manuscript overlaps significantly with the recent excellent review article of Moreira et al., 2018 published in Int. J. Mol. Sci.
During manuscript preparation we were not only aware of this recently published review, but also cited it in our manuscript. Essentially, the focus of the review by Moreira et al. and that of our manuscript are entirely different; we aimed to minimize overlap as much as possible, and this remains in our best interest.
The review of Moreira et al. focuses on pharmacological activities of pyrrolizidine alkaloids (PAs) and describes their anti-microbial, anti-inflammatory, anti-cancer and anti-viral potential in detail. In addition, the toxicity of PAs for humans and consumption including legal framework is extensively discussed.
In contrast, our manuscript focuses on biosynthesis and biological activities with a distinctive emphasis on both plant-herbivore interactions and occurrence of PAs in plants.
In the section concerning biochemistry, we provide a detailed summary of necine base formation and the enzymes involved in the process. Additionally, important stereochemical aspects are discussed. We also discuss the literature on the biosynthesis of necic acids including C5, C7, monocrotalic acid and senecic acid in detail. In contrast, Moreira et al. only briefly summarises those topics.
The following topic, regulation of PA levels by environmental cues, is not addressed at all in the manuscript of Moreira et al., 2018.
Overlap is also limited in section 5 of our manuscript. We initially discuss the role of PAs in plant ecology, mainly plant herbivore interactions. This topic is not addressed by Moreira et al., 2018.
The only significant overlap of the review of Moreira et al. and our manuscript can be found in the first part of section 5.2 (approximately one page), which discusses toxicity of PAs. However, we provide only a short summary of those topics while it is the main focus of the review of Moreira et al. Since we also aimed minimising overlap in that chapter we focussed in our manuscript more on detoxification mechanisms (pages 25 and 26), since that topic is not covered by Moreira et al.
The last part of our manuscript, occurrence of PAs in crop plants, is not covered by Moreira et al., 2018. Notably, for this section we focused explicitly on crop plants to avoid overlap with a number of reviews about occurrence of PAs in medicinal plants (Roeder 1995, 2000, 2009, 2011, 2015 and 2015 and Fu et al. 2009).
Yet another aspect of our manuscript worth consideration are the 277 references. From those only 27 (!) are also cited by Moreira et al., 2018. Thus, the overlap of the used literature is less than 10% and evidently minimal.
Taking the aforementioned reasons into consideration, there is no significant overlap of our manuscript with the review published by Moreira et al. In contrast, we think that both manuscripts complement each other in a comprehensive way, as they focus on different aspects of PAs.
However, we agree that the review of Moreira et al. should be explicitly mentioned in chapter 5.3 of our manuscript. We have included that in the revised manuscript (lines 891 to 894 of the revised manuscript).
In addition, we have deleted the part about legal aspects (lines 878-881 of the previous manuscript) since that topic is fully covered by the review of Moreira et al.
Round 2
Reviewer 3 Report
I have read the authors rebuttal with great care. In a general way, i agree with the authors.
The biosynthesis section and distribution in crops is mostly new and does not overlapp with previously published papers.
I advise the authors to remove or significantly cut the section regarding toxicology and biological activity as it has already been covered elsewhere less than 1 year ago.
Author Response
Response to Reviewer 3 Comments
Reviewer 3
I have read the authors rebuttal with great care. In a general way, i agree with the authors.
The biosynthesis section and distribution in crops is mostly new and does not overlapp with previously published papers.
Response:
We appreciate that reviewer 3 agrees that the main parts our manuscript are mostly new and do not overlap with previously published review articles.
Comment 1:
I advise the authors to remove or significantly cut the section regarding toxicology and biological activity as it has already been covered elsewhere less than 1 year ago.
Response 1:
We think that a review about pyrrolizidine alkaloids (PAs) must contain at least a brief summary about toxicity of PAs. Such a section is also necessary as a kind of introduction for the discussion of detoxification of PAs, a topic only briefly mentioned by Moreira et al. Thus, according to the recommendation of reviewer 3, we have further shortened chapter 5.3 by removing lines 905 to 915 and lines 956-965 (line numbers according to version R1).
Now, only a brief summary of less than one page about PA toxicity is given in the revised manuscript (lines 895 to 922; numbers according to version R2). Thus, we think that there is no substantial overlap to the review of Moreira at al.
